# AccidentBench: Benchmarking Multimodal Understanding and Reasoning in Vehicle Accidents and Beyond

## Abstract

Rapid advances in multimodal models demand benchmarks that rigorously evaluate understanding and reasoning in safety-critical, dynamic real-world settings. We present AccidentBench, a large-scale benchmark that combines *vehicle accident* scenarios with *Beyond* domains, safety-critical settings in air and water that emphasize spatial and temporal reasoning (e.g., navigation, orientation, multi-vehicle motion). The benchmark contains approximately 2,000 videos and over 19,000 human-annotated question–answer pairs spanning multiple video lengths (short/medium/long) and difficulty levels (easy/medium/hard). Tasks systematically probe core capabilities: temporal, spatial, and intent understanding and reasoning. By unifying accident-centric traffic scenes with broader safety-critical scenarios in air and water, AccidentBench offers a comprehensive, physically grounded testbed for evaluating models under real-world variability. Evaluations of state-of-the-art models (e.g., Gemini 2.5 Pro and GPT 5) show that even the strongest models achieve only about 18% accuracy on the hardest tasks and longest videos, revealing substantial gaps in real-world temporal, spatial, and intent reasoning. AccidentBench is designed to expose these critical gaps and drive the development of multimodal models that are safer, more robust, and better aligned with real-world safety-critical challenges. The code and dataset are available at: http://accident-bench.site

## 1 Introduction

As artificial intelligence (AI) continues to evolve, large multimodal models have shown impressive capabilities across vision, language, and video domains. However, significant challenges remain in deploying these models for real-world, safety-critical applications such as autonomous driving, robotics, and aerial or maritime operations. While multimodal models demonstrate remarkable performance in constrained or simulated environments, their robustness and depth of understanding in high-stakes, dynamic scenarios are still far from sufficient.

In particular, deployment in mission-critical domains requires rigorous evaluation of models' understanding and reasoning abilities under real-world conditions that involve uncertainty, physical interactions, and causal dependencies. While recent benchmarks have advanced evaluation in specific facets like temporal understanding (e.g., MVBench (Li et al., 2024c), REXTIME (Chen et al., 2024a)) or domain-specific knowledge (e.g., MMMU (Yue et al., 2024), DriveLM (Sima et al., 2024b)), there remains a paucity of unified platforms that assess understanding and reasoning across diverse vehicle accident and other open-space domains. To address this, we designed AccidentBench to rigorously evaluate multimodal models' understanding and reasoning in safety-critical tasks, with a primary focus on traffic accident scenarios and other high-stakes real-world settings.

Specifically, AccidentBench targets understanding and reasoning across diverse vehicle accident scenarios (83.0%), while also encompassing airspace (10.2%) and waterway (6.8%) domains, in which safety, perception, and decision-making are deeply interdependent. Unlike benchmarks that emphasize isolated skills or single domains, AccidentBench systematically challenges models across several critical understanding and reasoning capabilities: temporal understanding and reasoning (tracking event sequences and causality over extended periods); spatial understanding and reasoning

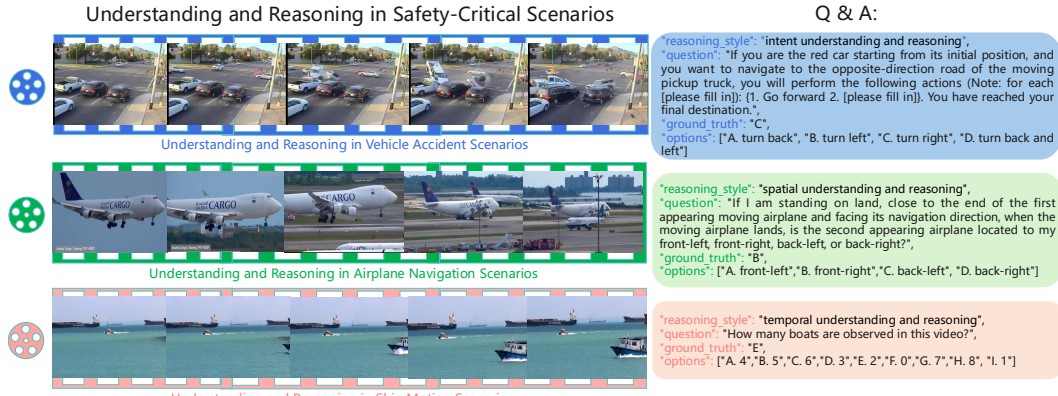

Figure 1: Examples of multimodal understanding and reasoning in vehicle accident and other safety-critical scenarios.

(understanding dynamic spatial relationships and multi-agent trajectories); and intent and goal reasoning (inferring agent intentions and planning goals), which further includes complex strategic and counterfactual reasoning (evaluating higher-order strategies, action implications, and "what-if" scenarios). Representative examples from AccidentBench are illustrated in Figure 1. By probing these abilities across diverse, safety-critical scenarios, AccidentBench offers a rigorous framework for assessing progress toward multimodal AI systems capable of reliable real-world operation.

Our key contributions are summarized as follows:

- **Vehicle Accident Focus:** We introduce AccidentBench, which emphasizes diverse vehicle accident scenarios while also extending to airspace and waterway domains. Evaluating vehicle accidents is especially critical for the safe deployment of LLMs in real-world applications and is a key step toward their widespread use in autonomous driving.

- **Real-World Limitations and Safety Gaps:** We highlight weaknesses in current AI systems' understanding and reasoning across open-space domains (e.g., autonomous driving, aviation, and marine) and provide a challenging testbed to advance safer and more reliable multimodal models.

- **Fine-grained reasoning evaluation:** AccidentBench systematically tests temporal, spatial, and intent reasoning across easy/medium/hard difficulties and short/medium/long video lengths, enabling precise diagnosis of model capabilities.

- **Unified Evaluation Suite with High-Quality Datasets:** AccidentBench provides human-labeled high-quality datasets and offers a video-based framework that integrates land traffic, airspace, and waterway scenarios, systematically evaluating temporal understanding, spatial understanding, and intent/goal reasoning within dynamic, safety-critical environments.

## 2 RELATED WORK

### 2.1 GENERAL MULTIMODAL UNDERSTANDING BENCHMARKS

Recent years have witnessed growing interest in video understanding benchmarks. Foundational video question-answering (QA) efforts include MSR-VTT (Xu et al., 2016) and Next-QA (Xiao et al., 2021). More recently, MVBench (Li et al., 2024c), with its 20 diverse temporal tasks derived from static images, and MLVU (Zhou et al., 2024a) have expanded video QA capabilities across multiple domains. The challenge of long-form video understanding has seen contributions from benchmarks such as EgoSchema (Mangalam et al., 2023), Video-LLaVA (Fu et al., 2024), MovieChat (Song et al., 2024), and LongVideoBench (Wu et al., 2024). Parallelly, video captioning benchmarks such as AuroraCap (Chai et al., 2024), HiCM2 (Kim et al., 2025), and LongCaptioning (Wei et al., 2025) focus on generating detailed textual descriptions.

A significant trend is the push for more rigorous temporal and causal reasoning. REXTIME (Chen et al., 2024a), for instance, probes the linking of causally related events across separate video

segments. For multi-domain understanding, MMWorld (He et al., 2025) evaluates models across diverse disciplines, requiring explanations and counterfactuals. Furthermore, LVBench (Wang et al., 2024) integrates video inputs for QA. Beyond video, reasoning from static images is explored by MME (Jiang et al., 2025) (including CoT extensions), MMMU (Yue et al., 2024) (evaluating expert-level multi-discipline reasoning), and benchmarks for mathematical reasoning like Dynamath (Zou et al., 2024) and MultiModal-MATH (Zhou et al., 2024b). For academic content, Video-MMLU (Song et al., 2025) offers a large-scale lecture video benchmark.

While these diverse benchmarks advance important aspects of multimodal understanding, such as general video comprehension, temporal analysis, long-form narrative understanding, captioning, and static image reasoning, they typically lack a unified framework for evaluation across land, air, and maritime open-space environments. Moreover, they may not capture the specific combination of complex reasoning skills, including strategic and intent-based inference, that AccidentBench is designed to assess in these contexts.

## 2.2 Safety-Critical Multimodal Understanding Benchmarks

Evaluating models in safety-critical domains, where understanding and reasoning under uncertainty is vital, is an emerging focus. Initial efforts addressed static image safety (Liu et al., 2024a), model robustness against adversarial attacks (e.g., FigStep (Gong et al., 2023), JailBreakV (Luo et al., 2024)) (Shayegani et al., 2023; Qi et al., 2024), or indoor robotics (Yang et al., 2024).

Autonomous driving has been a major driver of safety-critical research. Foundational datasets such as nuScenes[1] and Waymo Open Dataset[2], along with language-integrated efforts such as DriveLM and DriveVLM (Sima et al., 2024b; Tian et al., 2025), are closely related to AccidentBench's goals due to their real-world video and safety considerations. However, a key motivation for AccidentBench is that these traditionally emphasized perception and planning, with less focus on deep safety-critical reasoning for tasks such as accident cause analysis or complex decision-making. Other useful specialized benchmarks address related problems such as video anomaly detection, traffic-accident description, video-based reasoning, narration-driven road-event understanding, and audio-visual anomaly-understanding (e.g., VANE-Bench (Gani et al., 2025), AVD2 (Li et al., 2025), EchoTraffic (Xing et al., 2025b), RoadSocial (Parikh et al., 2025b), and SUTD-TrafficQA (Xu et al., 2021)). In contrast, our benchmark centers on safety-critical reasoning in accident and risk scenarios that require deep causal and counterfactual understanding, and introduces multi-level reasoning, temporal, spatial, and intent. Beyond traffic scenes, our dataset encompasses airplane navigation and ship motion scenarios, offering a multi-domain evaluation of spatial intelligence under dynamic and safety-critical conditions.

While advancements continue in specialized video reasoning and domain-specific safety evaluations, existing benchmarks still largely focus on single operational domains. Critically, they may lack sufficient coverage of high-risk scenarios such as traffic collisions, ship navigation, and airplane takeoff/landing events across combined land, air, and water settings. A unified platform to consistently evaluate robust, generalizable reasoning (e.g., temporal-causal, spatial, intent, and strategic analysis) across these diverse, safety-critical real-world scenarios also remains absent. To address this specific void, AccidentBench distinctively incorporates these challenging high-risk scenarios from all three domains. The reliability of its complex reasoning evaluation is ensured as all annotations were generated by highly educated annotators. AccidentBench thus provides a much-needed testbed for fostering robust, adaptable AI capable of safety-critical scenario understanding.

---

[1] https://www.nuscenes.org/
[2] https://waymo.com/open/

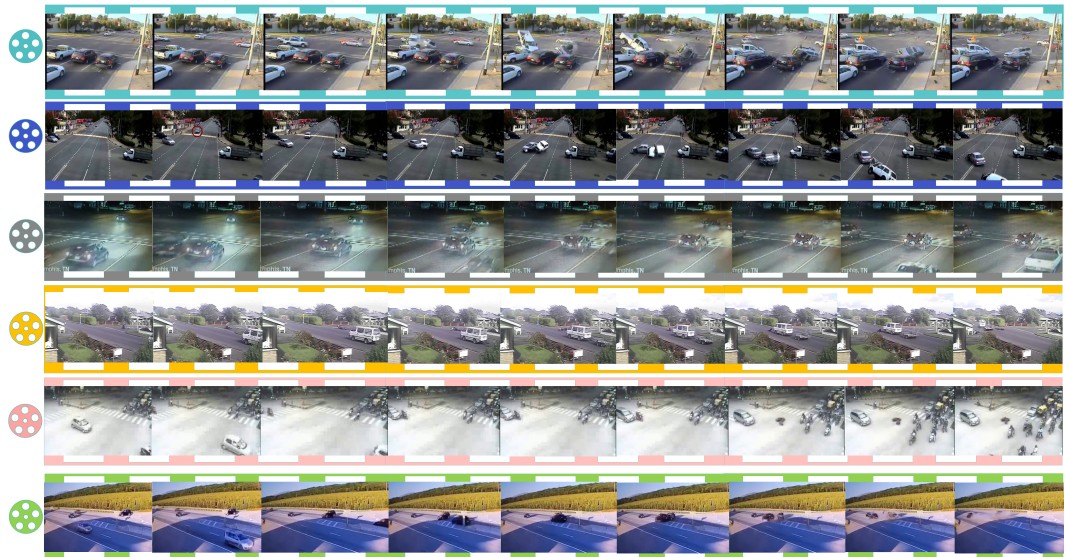

Figure 2: Land-space traffic accident scenarios for open-space video understanding and reasoning include intersection collisions, urban road accidents, nighttime incidents, rural road accidents, snow-covered road collisions, and freeway accidents.

Table 1: Overview of traffic accident scenarios in our benchmark, covering diverse road environments, weather conditions, and involved traffic participants.

| Index | Categories |
|---|---|
| **Road Environments:** | Intersection, Highway, Freeway, Rural Road, Tunnel, Urban Road, Bridge, Parking Lot |
| **Weather Conditions:** | Snow, Rain, Sunshine, Cloudy, Foggy, Windy |
| **Involved Participants:** | Sedan, SUV, Bus, Truck, Motorcycle, Bicycle, Van, Pickup, Trailer, Pedestrian |

## 3 BENCHMARK DESIGN AND ANALYSIS

### 3.1 SCENARIO SETTINGS

In this benchmark, we include diverse real-world scenario datasets [3], with a primary focus on traffic accident understanding and reasoning. Vehicle accident scenarios account for 83% of the dataset. In addition, we incorporate high-stakes, safety-critical settings such as airplane navigation scenarios, which account for 10.2% and focus on takeoff and landing, and ship motion scenarios, which account for 6.8% and emphasize navigation understanding and reasoning.

**Vehicle Accident Scenarios**  In the scenarios, we include a comprehensive range of traffic accident scenarios, encompassing diverse collision events under varying weather conditions such as snow, rain, and sunshine, as detailed in Table 1. Specific examples of these scenarios are illustrated in Figure 2, and more detailed examples are provided in Appendix F. To enhance contextual diversity, we incorporate multiple camera perspectives, including ego-centric and third-person views, particularly for accident scenes. The dataset features incidents involving a wide variety of vehicle types, including buses, motorcycles, sedans, and several categories of trucks, across different road environments such as highways, freeways, and rural roads. The associated questions are designed to evaluate models across multiple reasoning dimensions, including temporal-causal understanding, spatial reasoning, and intent and goal planning. The original video datasets are sourced from (Bao et al., 2020; Shah et al., 2018), which primarily collected videos from YouTube and other public internet platforms.

**Other Safety-Critical Scenarios**  *(1) Ship Motion Scenarios:* These scenarios include both **river** and **ocean** settings, covering diverse boats and ships under varying navigation conditions. These environments are critical yet underexplored in multimodal research. We assess temporal, spatial, and

---

[3]These datasets are used solely for academic research. They are employed only to evaluate model performance in this work.

| Temporal Understanding and Reasoning | Intent Understanding and Reasoning | Spatial Understanding and Reasoning |
|---|---|---|
| 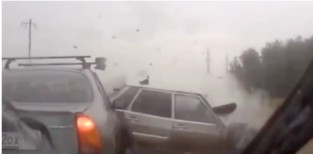 | 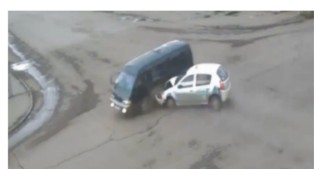 | 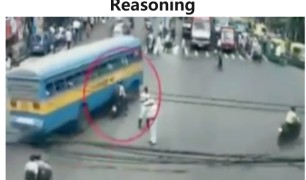 |
| "question": "There are multiple accidents, in the third accident, how many moving cars came from the opposite direction of the violated traffic rule car?" Options: A. 1, B. 2, C. 3, D. 4, E. 5, F. 6, G. 7, H. 8, I. 9, J. 10, K. 11, L. 12" | "question": "As the driver of the violated traffic rule car starting from its initial position, what sequence of actions would you take to avoid the collision? (Please fill in each step): {1. [ ] 2. [ ]} Options: A. Stop, B. Turn left, C. Reduce speed, D. Go forward, E. Turn right" | "question": "There are multiple accidents in the video, in the third accident, if I stand facing the initial direction of the blue collision bus near the driver at the moment of impact, is the red vehicle located front-left, front-right, back-left, or back-right relative to me?" |

Figure 3: Examples of question settings in AccidentBench across three key understanding and reasoning types: *Temporal Understanding and Reasoning*, which involves understanding event sequences and motion over time; *Spatial Understanding and Reasoning*, which focuses on relative positioning and orientation in space; and *Intent Understanding and Reasoning*, which evaluates understanding of goal-directed behaviors and decision-making in dynamic environments.

intent/goal understanding and reasoning through video-based tasks of different lengths and difficulty levels, using both interval-based and accuracy-based formats. The water-space videos are sourced from publicly available datasets, including (Guo et al., 2023; Prasad et al., 2017). *(2) Airplane Navigation Scenarios:* These scenarios primarily involve takeoff and landing events, emphasizing airplane navigation and perceptual understanding and reasoning. Despite their real-world importance, airplanes also remain underexplored in multimodal research. Our benchmark captures variations in navigation patterns, aircraft sizes, and motion dynamics across different airplane types. These scenarios include videos of varying lengths and evaluate models on spatial, temporal, and intent/goal understanding and reasoning across multiple difficulty levels using both interval-based and accuracy-based multiple-choice formats. The airspace videos are sourced from publicly available footage[4],[5],[6].

## 3.2 TASK SETTINGS

Within each scenario, we design tasks that evaluate models across three key dimensions of understanding and reasoning: *temporal*, *spatial*, and *intent and goal*. Representative examples for each type are shown in Figure 3.

For each understanding and reasoning dimension, we construct tasks at three difficulty levels using two formats: *interval-based choices* (easy and medium) and *accuracy-based choices* (hard). Easy tasks (≈6,300 QA pairs) provide approximately three coarse-grained interval options; medium tasks (≈6,300 QA pairs) include six intermediate-level intervals; and hard tasks (≈6,300 QA pairs) present fine-grained discrete options that require an exact match with the correct answer. Note that the primary differences between difficulty levels lie in the number and types of answer choices. The number of tasks is evenly distributed across difficulty levels, with each tier comprising one-third of the total. In all cases, the model must select a single best answer, allowing the benchmark to systematically assess performance under increasing levels of precision and ambiguity.

## 3.3 DATASET ANALYSIS

This benchmark includes approximately 2,000 videos and related massive human-annotated question-answer pairs, covering a wide range of understanding and reasoning tasks. The dataset features a variety of video lengths, categories, and frame counts, and spans real-world scenarios. An overview of the dataset's characteristics is provided in Appendix F, which illustrates the distributions of video duration, domain coverage, and task styles, along with details of the annotation procedure and difficulty levels.

---

[4]https://www.youtube.com/watch?v=i6CrbqeksJ8

[5]https://www.youtube.com/watch?v=k5yvzTw08K8

[6]https://www.youtube.com/watch?v=Bt9tpiAmTs8

Table 2: **Benchmark comparison** for multimodal understanding and reasoning tasks.

| Dataset | Safety | Traffic | Annotation | Real-World | Main Scenarios | # Video | Ave. Duration (s) | Question-answering Number | Type |
|---|---|---|---|---|---|---|---|---|---|
| MovieChat-1K (Song et al., 2023) | ✗ | ✗ | Human | ✓ | General | 1,000 | 564 | 13,000 | Open-ended |
| MMWorld (He et al., 2024) | ✗ | ✗ | Human | ✓ | General | 1,910 | 107 | 6,627 | Multiple-choice |
| MLVU (Zhou et al., 2024a) | ✗ | ✗ | Human | ✓ | General | 1,730 | 930 | 3,102 | Multiple-choice |
| MVBench (Abellán et al., 2023) | ✗ | ✗ | Human & LLM | ✓ | General | 4,000 | 16 | 4,000 | Multiple-choice |
| LongVideoBench (Wu et al., 2024) | ✗ | ✗ | Human | ✓ | General | 3,763 | 473 | 6,678 | Multiple-choice |
| TempCompass (Liu et al., 2024b) | ✗ | ✗ | Human & LLM | ✓ | General | 410 | < 30 | 7,540 | Multiple-choice |
| VSI-Bench (Yang et al., 2024) | ✗ | ✗ | Human | ✓ | Embodied | 288 | 50-100 | 5,000 | Multiple-choice |
| Video-MMMU (Hu et al., 2025) | ✗ | ✗ | Human & LLM | ✗ | Professional | 300 | 506 | 900 | Multiple-choice |
| Video-MMLU (Song et al., 2025) | ✗ | ✗ | Human & LLM | ✗ | Professional | 1,065 | 109 | 15,746 | Open-ended |
| DriveBench (Xie et al., 2025) | ✓ | ✓ | Human & LLM | ✓ | General Driving | ✗ | ✗ | 19,200 | Multiple-choice |
| DriveLM (Sima et al., 2024a) | ✓ | ✓ | Human | ✓ | General Driving | ✗ | ✗ | 15,480 | Open-ended |
| nuScenes-QA (Qian et al., 2024) | ✗ | ✓ | Human | ✓ | General Driving | ✗ | ✗ | 83,337 | Open-ended |
| MSSBench (Zhou et al., 2024b) | ✓ | ✗ | Human & LLM | ✓ | General | ✗ | ✗ | 1960 | Open-ended |
| MMSBench (Liu et al., 2024a) | ✓ | ✗ | LLM | ✓ | General | ✗ | ✗ | 5040 | Open-ended |
| RoadSocial (Parikh et al., 2025a) | ✓ | ✓ | LLM | ✓ | General Driving | 13.2k | 35.6 | 260k | Open-ended |
| Echotraffic (Xing et al., 2025a) | ✓ | ✓ | LLM | ✓ | General Driving | 29,865 | 3.76 | 149,325 | Open-ended |
| AVD2 (Li et al., 2025) | ✓ | ✓ | LLM | ✗ | General Driving | 8,399 | 63 | - | Open-ended |
| AccidentBench (ours) | ✓ | ✓ | Human | ✓ | Accident | 2000 | 56 | 19,000 | Multiple-choice |

Table 3: Understanding and reasoning evaluation for AccidentBench in Vehicle Accidents.

| Models | Size | Hard Avg. | Hard Temp. | Hard Spatial | Hard Intent | Medium Avg. | Medium Temp. | Medium Spatial | Medium Intent | Easy Avg. | Easy Temp. | Easy Spatial | Easy Intent |
|---|---|---|---|---|---|---|---|---|---|---|---|---|---|
| GPT 5 (OpenAI, 2025) | - | **37.33** | 35.85 | 42.80 | 33.35 | **48.34** | 46.22 | 55.07 | 43.74 | **54.86** | 52.35 | 55.50 | 56.72 |
| GPT 4o (Hurst et al., 2024) | - | 24.41 | 27.93 | 30.94 | 14.61 | 36.99 | 38.05 | 41.89 | 30.78 | 45.92 | 52.63 | 47.32 | 41.56 |
| Gemini 2.5 pro (Google, 2025b) | - | 29.76 | 30.58 | 34.42 | 24.27 | 36.46 | 35.32 | 43.96 | 30.11 | 54.56 | 54.31 | 56.35 | 53.03 |
| Gemini 2.5 flash think (Google, 2025a) | - | 28.67 | 26.30 | 38.27 | 21.42 | 37.52 | 34.07 | 47.41 | 31.33 | 50.00 | 51.31 | 48.17 | 50.53 |
| Gemini 2.5 flash no-think (Google, 2025a) | - | 24.34 | 32.54 | 24.58 | 15.89 | 36.70 | 37.36 | 42.96 | 29.78 | 51.40 | 53.65 | 49.21 | 51.33 |
| Gemini 1.5 pro (DeepMind, 2024) | - | 18.76 | 19.70 | 20.83 | 15.75 | 33.89 | 29.25 | 45.41 | 27.00 | 46.00 | 55.67 | 43.58 | 39.00 |
| Claude 3.5 (Anthropic, 2024) | - | 28.71 | 28.30 | 31.11 | 19.21 | 35.35 | 30.31 | 45.41 | 30.33 | 48.59 | 53.43 | 48.34 | 44.24 |
| InternVL2.5 (Chen et al., 2024b) | 26B | 23.78 | 29.33 | 29.00 | 13.00 | 35.11 | 39.67 | 42.00 | 23.67 | 52.55 | 60.67 | 55.00 | 42.00 |
| InternVL2.5 (Chen et al., 2024b) | 8B | 22.67 | 26.67 | 30.33 | 11.00 | 34.66 | 37.00 | 49.00 | 18.00 | 50.11 | 55.67 | 55.33 | 39.33 |
| InternVL2.5 (Chen et al., 2024b) | 4B | 19.56 | 20.00 | 24.67 | 13.33 | 33.89 | 32.67 | 42.33 | 26.67 | 44.89 | 46.00 | 48.67 | 40.00 |
| LLaVA Next (Li et al., 2024a) | 32B | 16.22 | 12.67 | 24.67 | 11.33 | 20.00 | 15.33 | 31.67 | 13.00 | 31.25 | 22.33 | 36.33 | 35.33 |
| LLaVA Video (Zhang et al., 2024b) | 7B | 19.78 | 16.00 | 31.00 | 12.33 | 25.67 | 23.33 | 34.00 | 21.33 | 31.44 | 28.00 | 33.00 | 33.33 |
| LLaVA OneVision (Li et al., 2024b) | 7B | 13.67 | 9.67 | 19.00 | 12.33 | 16.67 | 18.67 | 22.67 | 16.67 | 29.78 | 28.33 | 33.00 | 28.00 |
| Qwen2.5 VL (Bai et al., 2025) | 32B | 22.66 | 20.33 | 28.00 | 19.67 | 28.55 | 23.00 | 38.00 | 24.67 | 43.22 | 45.33 | 44.00 | 40.33 |
| Qwen2.5 VL (Bai et al., 2025) | 7B | 22.89 | 19.67 | 30.67 | 18.33 | 29.89 | 28.33 | 36.00 | 25.33 | 40.67 | 40.33 | 37.33 | 44.33 |
| Random Guess | - | 17.38 | 14.91 | 26.65 | 10.57 | 26.85 | 24.88 | 32.12 | 23.54 | 37.35 | 33.61 | 45.11 | 33.33 |

## 3.4 COMPARISON WITH EXISTING BENCHMARKS

Table 2 provides a comparative analysis of AccidentBench alongside existing evaluation benchmarks for multimodal models. Most benchmarks primarily focus on assessing the multimodal reasoning capabilities of multimodal models (He et al., 2024; Song et al., 2023; Zhou et al., 2024a); however, a significant limitation is the prevalent oversight of safety considerations. While a few recent benchmarks have begun to evaluate safety aspects of multimodal models (Zhou et al., 2024b; Liu et al., 2024a), they typically do not incorporate video-based question answering and are mostly limited to single-frame inputs. However, single-frame capture often introduces uncertainties in reasoning and is insufficient for reliably assessing multimodal models' ability to handle safety-critical issues. In contrast, our AccidentBench introduces a large-scale curated collection of video question-answer pairs that specifically focus on traffic accident understanding and reasoning in real-world safety-related scenarios. Comprising 2,000 carefully selected videos and 6,300 hard-level question–answer pairs, extended to medium and easy levels by varying the number of answer choices, AccidentBench includes a total of over 19,000 question–answer pairs. This scale is competitive with existing benchmarks and highlights the comprehensiveness of our evaluation set.

## 4 EXPERIMENTS

In our experiments, we build upon the `lmms-eval` framework (Zhang et al., 2024a) as the foundation for our benchmark and extend it to support the specific requirements of AccidentBench. We conduct comprehensive evaluations to assess the performance of state-of-the-art (SOTA) multimodal models across diverse safety-critical real-world scenarios.

## 4.1 EVALUATION IN VEHICLE ACCIDENT SCENARIOS

We evaluate model performance across all vehicle accident scenarios in AccidentBench, with results summarized in Table 3. The evaluation is organized by task difficulty (Easy, Medium, Hard) and

Table 4: Evaluation of AccidentBench on **vehicle accident** scenarios using **short**, **medium**, and **long** videos, categorized by reasoning types and based on a subset of the dataset. The choices are **accuracy-based**, corresponding to the hard setting.

| Difficulty | Models | Size | Over. Avg. | Short Video Scenarios | | | | Medium Video Scenarios | | | | Long Video Scenarios | | | |
|---|---|---|---|---|---|---|---|---|---|---|---|---|---|---|---|
| | | | | Avg. | Temporal | Spatial | Intent | Avg. | Temporal | Spatial | Intent | Avg. | Temporal | Spatial | Intent |
| Hard | GPT 5 (OpenAI, 2025) | - | **37.33** | **45.87** | 48.52 | 55.10 | 34.00 | **48.12** | 49.02 | 39.29 | 56.06 | 18.00 | 10.00 | 34.00 | 10.00 |
| | GPT 4o (Hurst et al., 2024) | - | 24.41 | 26.78 | 34.65 | 34.69 | 11.00 | 35.70 | 43.14 | 32.14 | 31.82 | 11.00 | 6.00 | 26.00 | 1.00 |
| | Gemini 2.5 pro (Google, 2025b) | - | 29.76 | 34.84 | 36.63 | 44.90 | 23.00 | 35.76 | 45.10 | 30.36 | 31.82 | **18.67** | 10.00 | 28.00 | 18.0 |
| | Gemini 2.5 flash think (Google, 2025a) | - | 28.67 | 32.13 | 35.64 | 37.75 | 23.00 | 35.20 | 37.25 | 41.07 | 27.27 | **18.67** | 6.00 | 36.00 | 14.00 |
| | Gemini 2.5 flash no-think (Google, 2025a) | - | 24.34 | 24.74 | 30.69 | 26.53 | 17.00 | 30.94 | 52.94 | 23.21 | 16.67 | 17.33 | 14.00 | 24.00 | 14.00 |
| | Gemini 1.5 pro (DeepMind, 2024) | - | 18.76 | 19.72 | 23.76 | 20.41 | 15.00 | 24.55 | 33.33 | 16.07 | 24.24 | 12.00 | 2.00 | 26.00 | 8.00 |
| | Claude 3.5 (Anthropic, 2024) | - | 28.71 | 33.76 | 35.64 | 31.63 | 34.00 | 28.87 | 37.26 | 35.71 | 13.63 | 16.00 | 12.00 | 26.00 | 10.0 |
| | InternVL2.5 (Chen et al., 2024b) | 26B | 23.78 | 21.33 | 26.00 | 31.00 | 7.00 | 32.00 | 46.00 | 32.00 | 18.00 | 18.00 | 16.00 | 24.00 | 14.00 |
| | InternVL2.5 (Chen et al., 2024b) | 8B | 22.67 | 20.00 | 18.00 | 33.00 | 9.00 | 30.00 | 46.00 | 30.00 | 14.00 | 18.00 | 16.00 | 28.00 | 10.00 |
| | InternVL2.5 (Chen et al., 2024b) | 4B | 19.56 | 18.67 | 18.00 | 28.00 | 8.00 | 28.00 | 34.00 | 24.00 | 26.00 | 12.00 | 8.00 | 22.00 | 6.00 |
| | LLaVA Next (Li et al., 2024a) | 32B | 16.22 | 20.67 | 16.00 | 32.00 | 14.00 | 11.33 | 12.00 | 12.00 | 10.00 | 16.67 | 10.00 | 30.00 | 10.00 |
| | LLaVA Video (Zhang et al., 2024b) | 7B | 19.78 | 19.33 | 12.00 | 35.00 | 11.00 | 24.67 | 26.00 | 30.00 | 18.00 | 15.33 | 10.00 | 28.00 | 8.00 |
| | LLaVA OneVision (Li et al., 2024b) | 7B | 13.67 | 14.33 | 5.00 | 27.00 | 11.00 | 14.67 | 18.00 | 8.00 | 18.00 | 12.00 | 6.00 | 22.00 | 8.00 |
| | Qwen2.5 VL (Bai et al., 2025) | 32B | 22.66 | 19.33 | 11.00 | 34.00 | 13.00 | 35.33 | 46.00 | 24.00 | 36.00 | 13.33 | 4.00 | 26.00 | 10.00 |
| | Qwen2.5 VL (Bai et al., 2025) | 7B | 22.89 | 26.00 | 17.00 | 30.00 | 31.00 | 30.00 | 40.00 | 32.00 | 18.00 | 12.67 | 2.00 | 30.00 | 6.00 |

reasoning type (Temporal, Spatial, Intent). Among the models, **GPT 5** achieves the strongest overall performance, leading in the Hard setting with an average score of 37.33 and maintaining high results in Medium (48.34). **Gemini 2.5 Pro** also performs consistently well, ranking good in the Easy setting (54.56) and remaining competitive in Medium and Hard. **GPT 4o** shows strong results in Medium and Easy tasks, particularly in temporal and spatial reasoning. *Across all models, performance declines substantially as task difficulty increases, with intent reasoning under the Hard setting posing the most difficult challenge.* Overall, proprietary models (e.g., GPT 5, Gemini, GPT 4o) outperform open-source counterparts, but none achieves robust performance across all difficulty levels and reasoning types.

### 4.2 VEHICLE ACCIDENT EVALUATION ANALYSIS

To investigate how video length and task format affect model performance in vehicle accident scenarios, we report results from accuracy-based (hard) experiments and interval-based (easy and medium) experiments across short, medium, and long video lengths.

**Accuracy-Based Settings** As shown in Table 4, we present a comprehensive evaluation of model performance in the **Vehicle Accident** scenarios of AccidentBench, categorized by task type, video length. In the hard (accuracy-based) setting, performance drops significantly across all models as video length increases. For example, in hard tasks involving long videos, *even the best-performing models fall below 40% average accuracy and only achieve around 18% accuracy on the hardest tasks and longest video scenarios.* These results highlight the limitations of current multimodal models in handling complex, long-horizon real-world understanding and reasoning—particularly for extended temporal sequences, fine-grained spatial relations, and intent understanding and reasoning.

**Interval-Based Settings** As shown in Table 5, in the easy and medium (interval-based) settings, **GPT 5** achieves the strongest overall performance, reaching 54.86% accuracy, followed closely by **Gemini 2.5 Pro** at 54.56%. Other proprietary models, such as Gemini 2.5 flash and GPT 4o, also perform competitively, with GPT 4o attaining 52.39% overall accuracy. Among open-source systems, InternVL2.5 (26B) is the best performer, with an overall accuracy of 52.55%. While models like Gemini 2.5 flash (with think mode) and GPT 4o achieve relatively strong results on medium-difficulty tasks (37.53% and 36.99%, respectively), *performance consistently declines as video length increases, highlighting the persistent challenges in achieving robust understanding and reasoning across diverse real-world scenarios.*

### 4.3 OTHER OPEN-SPACE EVALUATION

Beyond vehicle accident evaluation, we also assess models in other high-stakes, safety-critical scenarios (17%), including **ship motion** (6.8%) and **airplane navigation** (10.2%). **Evaluation in Ship Motion Scenarios:** Table 6 shows results for multimodal models in the Water Space domain of AccidentBench, categorized by task difficulty (Easy, Medium, Hard) and reasoning type (Temporal, Spatial, Intent). **GPT 5** achieves the highest overall performance, leading in Hard (38.36), Medium (51.80), and Easy (63.00) tasks. **Gemini 2.5 Pro** remains competitive, with strong results on Hard tasks (29.64) and particularly strong intent reasoning. **Gemini 2.5 flash with think** also performs well,

Table 5: Evaluation of AccidentBench on **vehicle accident** scenarios using **short**, **medium**, and **long** videos, categorized by reasoning types and based on a subset of the dataset. The tasks use **interval-based** choices, corresponding to the easy and medium settings depending on the number of options.

| Difficulty | Models | Size | Over. Avg. | Short Video Scenarios | | | | Medium Video Scenarios | | | | Long Video Scenarios | | | |
|---|---|---|---|---|---|---|---|---|---|---|---|---|---|---|---|
| | | | | Avg. | Temporal | Spatial | Intent | Avg. | Temporal | Spatial | Intent | Avg. | Temporal | Spatial | Intent |
| Medium | GPT 5 (OpenAI, 2025) | - | **48.34** | 62.55 | 64.65 | 67.00 | 56.00 | 46.48 | 50.00 | 42.22 | 47.22 | 36.00 | 24.00 | 56.00 | 28.00 |
| | GPT 4o (Hurst et al., 2024) | - | 36.99 | 45.49 | 48.48 | 55.00 | 33.00 | 33.89 | 41.67 | 26.67 | 33.33 | 31.33 | 24.00 | 44.00 | 26.00 |
| | Gemini 2.5 pro (Google, 2025b) | - | 36.46 | 42.79 | 38.38 | 59.00 | 31.00 | 33.93 | 39.58 | 28.89 | 33.33 | 32.67 | 28.00 | 44.00 | 26.0 |
| | Gemini 2.5 flash think (Google, 2025a) | - | 37.52 | 47.82 | 46.47 | 56.00 | 41.00 | 36.99 | 43.75 | 42.22 | 25.00 | 28.00 | 12.00 | 44.00 | 28.00 |
| | Gemini 2.5 flash no-think (Google, 2025a) | - | 36.70 | 47.50 | 48.49 | 58.00 | 36.00 | 33.93 | 39.58 | 28.89 | 33.33 | 28.67 | 24.00 | 42.00 | 20.00 |
| | Gemini 1.5 pro (DeepMind, 2024) | - | 33.89 | 39.47 | 42.42 | 42.00 | 34.00 | 33.52 | 33.33 | 42.22 | 25 | 28.67 | 12.00 | 52.00 | 22.00 |
| | Claude 3.5 (Anthropic, 2024) | - | 35.35 | 41.78 | 35.35 | 50.00 | 40.00 | 35.60 | 39.58 | 42.22 | 25.00 | 28.67 | 16.00 | 44.00 | 26.0 |
| | InternVL2.5 (Chen et al., 2024b) | 26B | 35.11 | 36.00 | 39.00 | 50.00 | 19.00 | 36.67 | 50.00 | 36.00 | 24.00 | 32.67 | 30.00 | 40.00 | 28.00 |
| | InternVL2.5 (Chen et al., 2024b) | 8B | 34.66 | 37.33 | 43.00 | 57.00 | 12.00 | 35.33 | 42.00 | 46.00 | 18.00 | 31.33 | 26.00 | 44.00 | 24.00 |
| | InternVL2.5 (Chen et al., 2024b) | 4B | 33.89 | 39.67 | 38.00 | 53.00 | 28.00 | 32.67 | 44.00 | 28.00 | 26.00 | 29.33 | 16.00 | 46.00 | 26.00 |
| | LLaVA Next (Li et al., 2024a) | 32B | 20.00 | 27.33 | 16.00 | 49.00 | 17.00 | 10.67 | 14.00 | 10.00 | 8.00 | 22.00 | 16.00 | 36.00 | 14.00 |
| | LLaVA Video (Zhang et al., 2024b) | 7B | 25.67 | 25.00 | 20.00 | 34.00 | 26.00 | 23.33 | 14.00 | 40.00 | 16.00 | 22.33 | 14.00 | 40.00 | 16.00 |
| | LLaVA OneVision (Li et al., 2024b) | 7B | 16.67 | 16.00 | 26.00 | 30.00 | 16.00 | 14.67 | 18.00 | 8.00 | 18.00 | 19.33 | 12.00 | 30.00 | 16.00 |
| | Qwen2.5 VL (Bai et al., 2025) | 32B | 28.55 | 28.33 | 21.00 | 44.00 | 20.00 | 33.33 | 40.00 | 30.00 | 30.00 | 24.00 | 8.00 | 40.00 | 24.00 |
| | Qwen2.5 VL (Bai et al., 2025) | 7B | 29.89 | 39.00 | 37.00 | 42.00 | 38.00 | 30.67 | 32.00 | 40.00 | 20.00 | 20.00 | 16.00 | 26.00 | 18.00 |
| Easy | GPT 5 (OpenAI, 2025) | - | **54.86** | 71.20 | 76.00 | 69.61 | 68.00 | 48.71 | 47.06 | 44.90 | 54.17 | 44.67 | 34.00 | 52.00 | 48.00 |
| | GPT 4o (Hurst et al., 2024) | - | 47.17 | 52.35 | 59.00 | 47.06 | 51.00 | 47.16 | 54.9 | 41.67 | 46.00 | 42.00 | 44.00 | 50.00 | 32.00 |
| | Gemini 2.5 pro (Google, 2025b) | - | 54.56 | 62.96 | 70.00 | 55.88 | 63.00 | 54.73 | 52.94 | 59.18 | 52.08 | 46.00 | 40.00 | 54.00 | 44.00 |
| | Gemini 2.5 flash think (Google, 2025a) | - | 50.00 | 67.56 | 69.00 | 65.69 | 68.00 | 44.45 | 52.94 | 40.82 | 39.58 | 38.00 | 32.00 | 38.00 | 44.00 |
| | Gemini 2.5 flash no-think (Google, 2025a) | - | 51.40 | 58.97 | 70.00 | 54.90 | 52.00 | 46.56 | 52.94 | 36.74 | 50.00 | 48.67 | 38.00 | 56.00 | 52.00 |
| | Gemini 1.5 pro (DeepMind, 2024) | - | 46.00 | 51.33 | 60.00 | 50.00 | 44.00 | 36.92 | 49.02 | 36.73 | 25.00 | 50.00 | 58.00 | 44.00 | 48.00 |
| | Claude 3.5 (Anthropic, 2024) | - | 48.59 | 60.33 | 61.00 | 50.00 | 70.00 | 35.29 | 35.29 | 51.02 | 22.73 | 49.33 | 64.00 | 44.00 | 40.0 |
| | InternVL2.5 (Chen et al., 2024b) | 26B | 52.55 | 61.00 | 62.00 | 59.00 | 62.00 | 45.33 | 58.00 | 44.00 | 34.00 | 51.33 | 62.00 | 62.00 | 30.00 |
| | InternVL2.5 (Chen et al., 2024b) | 8B | 50.11 | 55.67 | 55.00 | 60.00 | 52.00 | 44.67 | 58.00 | 42.00 | 34.00 | 50.00 | 54.00 | 64.00 | 32.00 |
| | InternVL2.5 (Chen et al., 2024b) | 4B | 44.89 | 53.33 | 46.00 | 60.00 | 54.00 | 37.33 | 48.00 | 38.00 | 26.00 | 44.00 | 44.00 | 48.00 | 40.00 |
| | LLaVA Next (Li et al., 2024a) | 32B | 31.25 | 38.00 | 35.00 | 45.00 | 34.00 | 21.33 | 12.00 | 14.00 | 38.00 | 34.67 | 20.00 | 50.00 | 34.00 |
| | LLaVA Video (Zhang et al., 2024b) | 7B | 31.44 | 33.00 | 30.00 | 31.00 | 38.00 | 33.33 | 38.00 | 36.00 | 26.00 | 28.00 | 16.00 | 32.00 | 36.00 |
| | LLaVA OneVision (Li et al., 2024b) | 7B | 29.78 | 32.00 | 31.00 | 33.00 | 32.00 | 24.00 | 26.00 | 30.00 | 16.00 | 33.33 | 28.00 | 36.00 | 36.00 |
| | Qwen2.5 VL (Bai et al., 2025) | 32B | 43.22 | 51.00 | 58.00 | 50.00 | 45.00 | 41.33 | 46.00 | 38.00 | 40.00 | 37.33 | 32.00 | 44.00 | 36.00 |
| | Qwen2.5 VL (Bai et al., 2025) | 7B | 40.67 | 51.33 | 55.00 | 42.00 | 57.00 | 36.00 | 32.00 | 42.00 | 34.00 | 34.67 | 34.00 | 28.00 | 42.00 |

Table 6: Understanding and reasoning evaluation for AccidentBench in ship motion scenarios.

| Models | Size | Hard | | | | Medium | | | | Easy | | | |
|---|---|---|---|---|---|---|---|---|---|---|---|---|---|
| | | Avg. | Temp. | Spatial | Intent | Avg. | Temp. | Spatial | Intent | Avg. | Temp. | Spatial | Intent |
| GPT 5 (OpenAI, 2025) | - | **38.36** | 38.08 | 31.38 | 45.62 | **51.80** | 54.92 | 47.08 | 53.38 | **63.00** | 69.77 | 49.00 | 70.23 |
| GPT 4o (Hurst et al., 2024) | - | 22.10 | 28.23 | 22.46 | 15.62 | 38.49 | 43.00 | 50.92 | 21.54 | 50.51 | 61.84 | 42.00 | 47.69 |
| Gemini 2.5 pro (Google, 2025b) | - | 29.64 | 30.54 | 25.31 | 33.08 | 41.77 | 39.38 | 53.77 | 32.16 | 61.05 | 64.84 | 53.84 | 64.46 |
| Gemini 2.5 flash think (Google, 2025a) | - | 27.36 | 30.38 | 24.46 | 27.23 | 46.77 | 54.9 | 49.85 | 38.16 | 62.03 | 75.38 | 47.16 | 63.54 |
| Gemini 2.5 flash no-think (Google, 2025a) | - | 27.44 | 39.16 | 19.62 | 23.54 | 46.12 | 51.08 | 50.84 | 36.08 | 58.18 | 68.84 | 40.23 | 65.46 |
| Gemini 1.5 pro (DeepMind, 2024) | - | 26.02 | 28.54 | 25.85 | 23.66 | 46.31 | 40.08 | 57.20 | 41.66 | 50.69 | 46.16 | 53.98 | 51.92 |
| Claude 3.5 (Anthropic, 2024) | - | 25.44 | 22.62 | 20.62 | 33.08 | 38.31 | 52.00 | 25.54 | 39.23 | 49.39 | 56.00 | 52.92 | 39.23 |
| InternVL2.5 (Chen et al., 2024b) | 26B | 22.54 | 16.69 | 23.62 | 27.31 | 41.77 | 27.38 | 59.84 | 38.08 | 55.05 | 57.69 | 53.84 | 53.62 |
| InternVL2.5 (Chen et al., 2024b) | 8B | 21.90 | 11.85 | 27.46 | 26.38 | 41.08 | 34.31 | 60.77 | 28.16 | 53.47 | 57.62 | 50.08 | 52.69 |
| InternVL2.5 (Chen et al., 2024b) | 4B | 20.92 | 17.62 | 22.62 | 22.54 | 44.36 | 25.54 | 62.69 | 44.84 | 53.87 | 52.92 | 56.84 | 51.84 |
| LLaVA Next (Li et al., 2024a) | 32B | 14.39 | 7.85 | 24.62 | 7.85 | 20.88 | 10.77 | 34.23 | 17.62 | 35.59 | 28.46 | 45.92 | 32.38 |
| LLaVA Video (Zhang et al., 2024b) | 7B | 14.00 | 11.69 | 21.54 | 8.77 | 21.92 | 19.62 | 28.46 | 17.69 | 31.03 | 26.38 | 36.31 | 30.38 |
| LLaVA OneVision (Li et al., 2024b) | 7B | 15.67 | 9.77 | 27.46 | 9.77 | 22.54 | 16.62 | 32.38 | 18.62 | 33.00 | 31.31 | 36.31 | 31.38 |
| Qwen2.5 VL (Bai et al., 2025) | 32B | 13.39 | 7.85 | 23.54 | 8.77 | 33.31 | 19.62 | 50.00 | 30.31 | 52.77 | 46.92 | 57.77 | 53.62 |
| Qwen2.5 VL (Bai et al., 2025) | 7B | 14.67 | 6.85 | 27.38 | 9.77 | 24.08 | 18.62 | 28.38 | 25.23 | 31.31 | 37.23 | 20.62 | 36.08 |

achieving good results among proprietary models in Medium and Easy settings. Among open-source models, InternVL2.5 (26B) model shows competitive performance, especially in intent reasoning, but still lag behind proprietary models. As with other domains, all models suffer a marked drop in performance on Hard tasks, most notably in intent reasoning. These findings emphasize the continued difficulty of multimodal reasoning in dynamic and ambiguous environments such as rivers and oceans, highlighting the need for more advanced AI systems. Due to space constraints, further analysis of ship motion across different video lengths and task modes, as well as the **Evaluation of Airplane Navigation Scenarios**, is provided in Appendix D.

These findings demonstrate AccidentBench's ability to reveal the limitations of existing multimodal models, particularly in safety-critical and physically grounded domains. *By highlighting domain-specific understanding and reasoning gaps, especially in underexplored high-stakes environments such as ship motion, and airplane navigation,* AccidentBench serves as a useful tool for guiding the development of more robust, spatially, temporally aware, and intent-aware multimodal systems.

## 4.4 MODEL ERROR ANALYSIS

To demonstrate the effectiveness of our benchmark and evaluate the performance of SOTA models, we conduct a qualitative analysis of model predictions on the AccidentBench benchmark. As shown

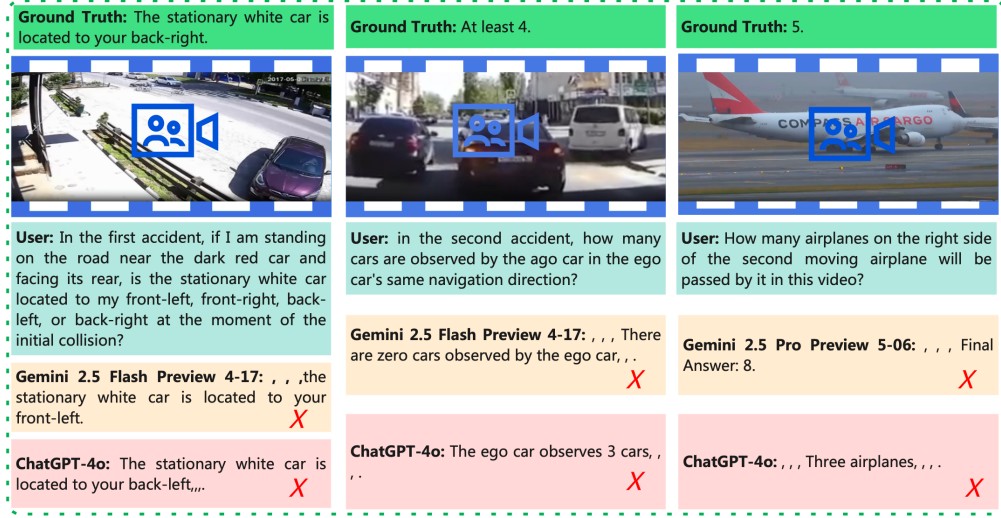

Figure 4: Qualitative error analysis of SOTA multimodal models (Gemini 2.5 and GPT 4o) on the AccidentBench benchmark. Each example illustrates a failure case in a different reasoning category: spatial reasoning (left), temporal reasoning (middle), and intent reasoning (right). Despite their capabilities, both models struggle with spatial localization, counting dynamic objects, and understanding goal-directed motion in real-world safety-critical scenarios.

in Figure 4, the analysis highlights persistent challenges in spatial, temporal, and intent understanding and reasoning across real-world environments. Despite the strong overall performance of leading multimodal models such as Gemini 2.5 and GPT 4o, the results reveal consistent failure cases in real-world scenarios. For example, both models struggle with accurately identifying spatial relationships (e.g., relative positions of vehicles), counting dynamic objects over time (e.g., cars in motion), and understanding goal-directed interactions (e.g., airplane passing events). *These failure cases highlight the limitations of current models in handling safety-critical, perception-intensive tasks.* By providing richly annotated, video-based tasks that demand multi-step reasoning grounded in physics, causality, and spatial understanding, AccidentBench serves as a rigorous diagnostic benchmark. Our findings highlight the necessity of such benchmarks for advancing the robustness, safety, and real-world applicability of large multimodal systems.

## 4.5 ABLATION EXPERIMENTS

In our experiments, due to the high cost of evaluating all data points, we adopt a uniform sampling strategy to select a representative subset of tasks. Specifically, for each understanding and reasoning type, we sample 50 tasks when the total number of available tasks is fewer than 500, and 100 tasks when the number exceeds 500. The AccidentBench spans three real-world scenarios, vehicle accident, airplane navigation, and ship motion, each with three video lengths (short, medium, long), three difficulty levels (easy, medium, hard), and three understanding and reasoning types: temporal, spatial, and intent-based understanding and reasoning. Following this sampling strategy, we evaluate a total of 3,798 tasks, evenly distributed across the three types: 1,266 *spatial understanding and reasoning*, 1,266 *temporal-causal understanding and reasoning*, and 1,266 *intent understanding and reasoning* tasks. To assess the reliability of this sampling approach, we conduct an ablation study comparing model performance on sampled tasks versus the full set of data points in the **vehicle accident (short, easy)** settings. We use InternVL 2.5, one of the leading open-source multimodal models, which ranks highly on several leaderboards such as [7] and [8]. As shown in Table 7, performance on the sampled subset is comparable to, and in some cases slightly better than, performance on the full dataset. These results validate the effectiveness of our sampling strategy in preserving benchmark consistency while reducing evaluation cost.

---

[7]https://enxinsong.com/Video-MMLU-web/
[8]https://huggingface.co/spaces/opencompass/open_vlm_leaderboard

Table 7: Performance comparison on **vehicle accident short videos** (easy setting): full vs. sampled data points.

| Model | Full Data Points | | | | Sample Data Points | | | |
|---|---|---|---|---|---|---|---|---|
| | Avg. | Temporal | Spatial | Intent | Avg. | Temporal | Spatial | Intent |
| InternVL2_5-26B | 55.62 | 57.61 | 50.37 | 58.88 | 61.00 | 62.00 | 59.00 | 62.00 |
| InternVL2_5-8B | 49.26 | 51.89 | 48.57 | 47.31 | 55.67 | 55.00 | 60.00 | 52.00 |
| InternVL2_5-4B | 50.65 | 50.17 | 50.70 | 51.10 | 55.33 | 52.00 | 55.00 | 59.00 |

## 5 CONCLUSION

In this work, we introduce AccidentBench, a large-scale benchmark for evaluating multimodal understanding and reasoning in real-world safety-scitical environments. AccidentBench provides richly annotated, video-based tasks designed to assess model performance across three fundamental understanding and reasoning dimensions: temporal, spatial, and intent and goal reasoning. The benchmark encompasses a broad range of scenarios, video lengths, and difficulty levels, enabling comprehensive evaluation in safety-critical, perception-intensive settings. Through extensive qualitative and quantitative analyses, we demonstrate that even SOTA multimodal models, both proprietary systems such as Gemini 2.5 Pro and GPT 5, and leading open-source models like Qwen and InternVL, exhibit significant limitations when understanding and reasoning over complex, dynamic physical environments. We hope that AccidentBench will serve as a valuable resource for the research community and help advance the development of safer, more generalizable, and practically deployable multimodal AI systems.

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

# Appendix

## A    LIMITATION AND IMPACT

**Limitation**    Our benchmark provides a valuable tool for evaluating model performance in safety-critical environments. However, due to the large scale of the dataset, evaluating all data points is computationally expensive. As a result, we were unable to perform large-scale testing with many high-cost proprietary models such as ChatGPT and Gemini. In future work, we plan to explore more efficient evaluation strategies and extend our analysis to a broader set of models, including closed-source systems.

**Impact**    This benchmark offers a new direction for advancing multimodal model development in open-space, safety-critical, and physically grounded real-world environments. By emphasizing temporal, spatial, and intent-based reasoning in diverse video scenarios, this benchmark can be useful to guide the design of more robust and reliable multimodal systems. While this research seeks to advance the capabilities of AI in complex settings, we do not identify any specific societal risks or consequences requiring special attention at this time.

## B    ERROR ANALYSIS

To better analyze model errors, we conducted additional experiments using the Gemini 2.5 Flash Reasoning model on 291 sampled reasoning traces. We examined the relationship between reasoning length and accuracy, as well as performance across different reasoning styles and difficulty levels.

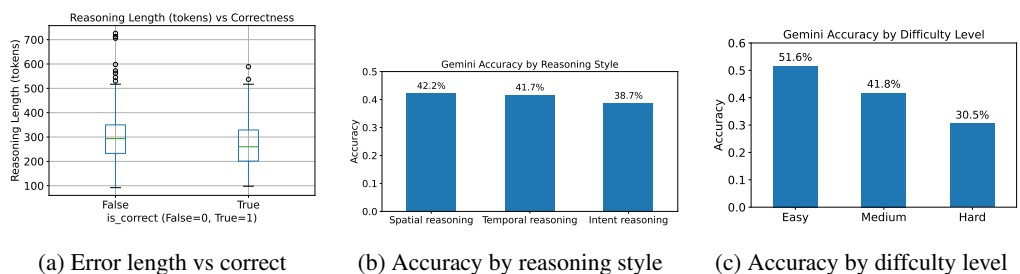

| (a) Error length vs correct | (b) Accuracy by reasoning style | (c) Accuracy by diffculty level |

Figure 5: Analysis of Gemini-2.5 Flash reasoning performance. (a) Relationship between reasoning-chain length and correctness. (b) Accuracy across different reasoning styles (spatial, temporal, intent). (c) Accuracy across difficulty levels (easy, medium, hard).

**(a) Reasoning length vs. correctness**    Figure 5(a) shows that incorrect predictions tend to be associated with longer reasoning chains. While the median reasoning length for correct answers is moderately shorter, the incorrect answers contain significantly more extreme outliers, with many explanations exceeding 500–700 tokens. This pattern indicates that verbose or highly elaborate explanations do not necessarily reflect stronger understanding, and in many cases correspond to hallucinated or compensatory reasoning when the model is uncertain. The findings align with prior observations that larger chain-of-thought does not guarantee improved factual accuracy.

**(b) Accuracy by reasoning style**    Figure 5(b) breaks down performance across reasoning types. Spatial reasoning achieves the highest accuracy (42%), followed closely by temporal reasoning (41%), while intent-goal reasoning is noticeably weaker (39%). This gap suggests that Gemini handles objective, visually grounded questions more reliably than questions requiring inference of agent intent or latent goals, which demand a deeper causal understanding of the scene. The relatively small separation between styles, however, also reflects that none of the reasoning categories are robust, and the model struggles across reasoning dimensions.

**(c) Accuracy by difficulty level** Figure 5(c) reveals a clear monotonic decline in performance as difficulty increases: easy scenarios reach over 51% accuracy, medium scenarios drop to 42%, and hard scenarios fall further to 30%. This demonstrates that Gemini's reasoning reliability is highly sensitive to task complexity. The steep degradation highlights a fundamental limitation: the model's reasoning does not generalize well to challenging open-world accident scenarios, even when the chain-of-thought appears detailed or structured.

## C  DIFFERENT CATEGORY ACCURACY: CASE STUDY

To further understand how environmental factors influence model performance in safety-critical video reasoning, we conduct a case study over a subset of evaluation scenes spanning diverse weather and geographic contexts. The left panel of Figure 6 shows the results across geographic settings. The model performs slightly better in rural-road scenarios (17.9%) than in urban-road scenarios (16.9%). This gap suggests that higher visual clutter, denser traffic patterns, and more heterogeneous object interactions in urban scenes pose additional challenges, causing the model to struggle with spatial and temporal grounding.

The right panel of Figure 6 demonstrates a more apparent difference across weather conditions. Under good-weather environments (21.0%), the model achieves substantially higher accuracy compared to bad-weather conditions (15.7%). Poor visibility caused by snow, rain, nighttime lighting, and other adverse conditions likely reduces visual signal quality and increases uncertainty in tracking, motion prediction, and causal reasoning. These results highlight the importance of environmental robustness and emphasize the need for training and evaluation pipelines that explicitly incorporate difficult weather and geographic distributions.

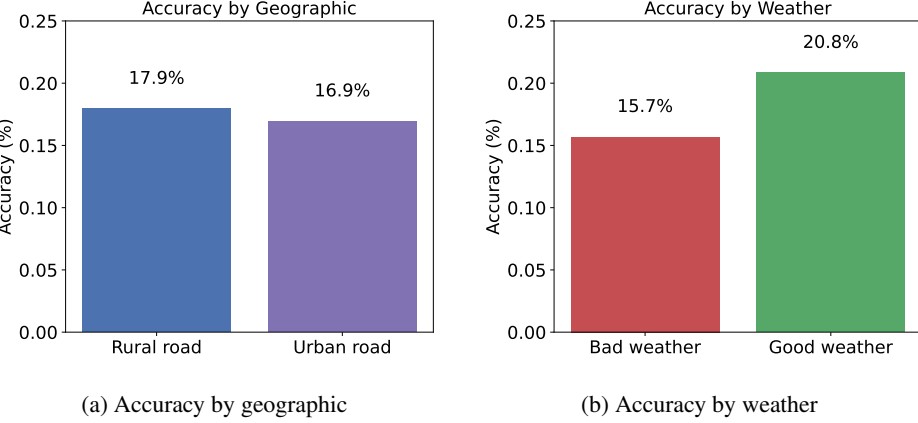

(a) Accuracy by geographic                    (b) Accuracy by weather

Figure 6: Model accuracy across different environmental categories. (a) Accuracy by geographic setting, comparing rural-road and urban-road scenarios. (b) Accuracy by weather conditions, comparing good-weather and bad-weather scenarios. Values denote average accuracy across all reasoning types.

## D  AIR SPACE EVALUATION:

Table 8 reports the evaluation results for multimodal models in the airplane navigation of AccidentBench. The results are broken down by task difficulty (Easy, Medium, Hard) and reasoning types (Temporal, Spatial, Intent). Gemini 2.5 Pro stands out with the strongest overall performance, achieving the highest average scores in hard tasks (31.39). GPT 5 and Gemini 2.5 flash think also perform competitively, for example, GPT 5 achieves good results on Easy tasks (52.00) and medium tasks (44.00), although it lags behind Gemini on harder examples. Open-source models such as InternVL2.5 models show moderate success in temporal reasoning but consistently underperform in intent reasoning. Overall, the trend mirrors that of the Land domain: performance declines signifi-

cantly as difficulty increases, with the largest drop occurring in temporal and intent reasoning tasks. These results emphasize the challenges multimodal models face in reliably operating in dynamic, real-world Air Space scenarios.

Table 8: Understanding and reasoning evaluation for AccidentBench in Airplane Navigation domain.

| Models | Size | Hard | | | | Medium | | | | Easy | | | |
|---|---|---|---|---|---|---|---|---|---|---|---|---|---|
| | | Avg. | Temp. | Spatial | Intent | Avg. | Temp. | Spatial | Intent | Avg. | Temp. | Spatial | Intent |
| GPT 5 (OpenAI, 2025) | - | 28.11 | 26.67 | 28.00 | 29.67 | 44.00 | 43.00 | 44.33 | 44.67 | 52.00 | 51.00 | 41.00 | 64.00 |
| GPT 4o (Hurst et al., 2024) | - | 18.11 | 11.00 | 30.33 | 13.00 | 38.45 | 47.00 | 40.00 | 28.33 | 40.67 | 33.00 | 38.67 | 50.33 |
| Gemini 2.5 pro (Google, 2025b) | - | **31.39** | 32.67 | 20.50 | 41.00 | 43.11 | 45.33 | 36.33 | 47.67 | **52.56** | 58.33 | 39.67 | 59.67 |
| Gemini 2.5 flash think (Google, 2025a) | - | 25.78 | 28.00 | 15.33 | 34.00 | 39.78 | 43.67 | 27.00 | 48.67 | 50.67 | 47.00 | 43.33 | 61.67 |
| Gemini 2.5 flash no-think (Google, 2025a) | - | 25.44 | 16.00 | 32.00 | 28.33 | **49.67** | 42.67 | 54.33 | 52.00 | 50.78 | 45.33 | 50.67 | 56.33 |
| Gemini 1.5 pro (DeepMind, 2024) | - | 22.34 | 18.00 | 24.33 | 24.67 | 38.78 | 32.00 | 51.67 | 32.67 | 43.00 | 39.67 | 42.00 | 47.33 |
| Claude 3.5 (Anthropic, 2024) | - | 24.22 | 16.00 | 33.33 | 23.33 | 39.67 | 36.00 | 43.00 | 40.00 | 42.45 | 34.67 | 46.33 | 46.33 |
| InternVL2.5 (Chen et al., 2024b) | 26B | 17.33 | 16.67 | 22.67 | 13.33 | 28.67 | 21.67 | 51.00 | 13.33 | 36.11 | 31.33 | 43.33 | 33.67 |
| InternVL2.5 (Chen et al., 2024b) | 8B | 18.22 | 13.67 | 31.00 | 10.00 | 34.33 | 30.67 | 51.00 | 21.33 | 38.44 | 40.00 | 42.67 | 32.67 |
| InternVL2.5 (Chen et al., 2024b) | 4B | 15.33 | 15.00 | 19.33 | 11.67 | 32.22 | 31.00 | 46.00 | 19.67 | 40.33 | 35.67 | 51.33 | 34.00 |
| LLaVA Next (Li et al., 2024a) | 32B | 17.89 | 8.33 | 35.33 | 10.00 | 26.11 | 20.33 | 40.00 | 18.00 | 33.22 | 35.67 | 34.67 | 29.33 |
| LLaVA Video (Zhang et al., 2024b) | 7B | 14.78 | 8.33 | 26.67 | 9.33 | 24.00 | 18.33 | 35.67 | 18.00 | 33.22 | 34.33 | 35.33 | 30.00 |
| LLaVA OneVision (Li et al., 2024b) | 7B | 15.67 | 11.33 | 26.33 | 9.33 | 23.67 | 20.00 | 33.67 | 17.33 | 33.22 | 34.33 | 35.33 | 30.00 |
| Qwen2.5 VL (Bai et al., 2025) | 32B | 16.22 | 3.33 | 30.00 | 15.33 | 33.34 | 18.00 | 52.67 | 29.33 | 52.45 | 43.00 | 56.67 | 57.67 |
| Qwen2.5 VL (Bai et al., 2025) | 7B | 16.55 | 2.33 | 30.00 | 17.33 | 28.00 | 25.00 | 23.33 | 35.67 | 39.89 | 43.67 | 21.33 | 54.67 |

# E   AIR AND WATER SPACE ANALYSIS:

Table 9 presents model performance in the **Airplane Navigation** of AccidentBench, evaluated across short, medium, and long video scenarios, and categorized by temporal, spatial, and intent reasoning tasks. In the easy setting, **Gemini 2.5 Pro** achieves the highest overall accuracy (52.56%), outperforming all other models, including GPT 4o and GPT 5. In the medium setting, Gemini 2.5 flash without think mode leads with 49.67%, followed closely by GPT 5 (44.00%) and Gemini Pro(43.11%). For hard tasks, which are the most challenging, **Gemini 2.5 Pro** remains the top performer with 31.39%. These results highlight the ability of the Gemini family of models to maintain performance in complex, dynamic airspace environments, but exhibit notable drops as the reasoning complexity increases, revealing current limitations in handling temporal, spatial, and intent-based challenges in aerial domains. Moreover, Table 10 presents model performance on the AccidentBench benchmark in the **Ship Motion**, covering both river and ocean scenarios across varying reasoning types and difficulty levels. GPT 5 model consistently outperforms other models across all settings.

# F   ANNOTATION AND DETAILED EXAMPLES

During data annotation, we first define the question types, then watch each video to design corresponding questions and annotate the answers. We first design the hard-level tasks and label each question with the ground-truth answer. Based on these, we then construct the medium and easy tasks. The primary differences between difficulty levels lie in the number and types of answer choices. Each question associated with a video is adjusted with different answer-choice configurations depending on the difficulty level. Our dataset contains approximately 2,101 videos and 19,136 question–answer pairs, evenly distributed across three difficulty levels: easy ($\approx$ 6,300 Q&A pairs), medium ($\approx$ 6,300 Q&A pairs), and hard ($\approx$ 6,300 Q&A pairs). The difficulty is determined by both the number and type of answer choices. Hard questions typically include 12 choices for temporal and intent reasoning, and 4 for spatial reasoning, requiring precise selection. Medium questions generally offer 6 choices for temporal and intent reasoning, and 3 for spatial reasoning, often involving interval-based options. Easy questions usually present 3 choices, or 2 for spatial reasoning, and also rely on interval-based distinctions.

Moreover, as illustrated in Figure 7, we present a detailed question-and-answer example. For each scenario's understanding and reasoning setting, we include three video lengths, short, medium, and long, each featuring tasks designed to evaluate temporal, spatial, and intent reasoning.

Specifically, as shown in Figure 8, in **(a) Video Length:** A substantial portion of the videos (76.5%) are short, with durations under 10 seconds. The remaining videos are distributed across longer intervals: 10–30 seconds (3.7%), 30–60 seconds (4.6%), 60–120 seconds (4.8%), 120–300 seconds (4.4%), and over 300 seconds (6.0%). This distribution reflects a strong emphasis on short,

Table 9: Evaluation of AccidentBench in the **Airplane Navigation** domain using **Short**, **Medium**, and **Long** Videos, categorized by reasoning types, based on a subset of the dataset.

| Difficulty | Models | Size | Over. Avg. | Short Video Scenarios | | | | Medium Video Scenarios | | | | Long Video Scenarios | | | |
|---|---|---|---|---|---|---|---|---|---|---|---|---|---|---|---|
| | | | | Avg. | Temporal | Spatial | Intent | Avg. | Temporal | Spatial | Intent | Avg. | Temporal | Spatial | Intent |
| Hard | GPT 5 (OpenAI, 2025) | - | 28.11 | 26.67 | 18.00 | 30.00 | 32.00 | 26.00 | 32.00 | 34.00 | 12.00 | 31.67 | 30.00 | 20.00 | 45.00 |
| | GPT 4o (Hurst et al., 2024) | - | 18.11 | 21.33 | 16.00 | 26.00 | 22.00 | 14.67 | 12.00 | 30.00 | 2.00 | 18.33 | 5.00 | 35.00 | 15.00 |
| | Gemini 2.5 pro (Google, 2025b) | - | **31.39** | 32.83 | 36.00 | 24.49 | 38.00 | 24.67 | 32.00 | 22.00 | 20.00 | 36.67 | 30.00 | 15.00 | 65.00 |
| | Gemini 2.5 flash think (Google, 2025a) | - | 25.78 | 26.00 | 26.00 | 18.00 | 34.00 | 21.33 | 28.00 | 18.00 | 18.00 | 30.00 | 30.00 | 10.00 | 50.00 |
| | Gemini 2.5 flash no-think (Google, 2025a) | - | 25.44 | 25.33 | 22.00 | 28.00 | 26.00 | 26.00 | 26.00 | 28.00 | 24.00 | 25.00 | 0.00 | 40.00 | 35.00 |
| | Gemini 1.5 pro (DeepMind, 2024) | - | 22.34 | 26.67 | 24.00 | 26.00 | 30.00 | 18.67 | 20.00 | 22.00 | 14.00 | 21.67 | 10.00 | 25.00 | 30.00 |
| | Claude 3.5 (Anthropic, 2024) | - | 24.22 | 26.00 | 18.00 | 32.00 | 28.00 | 23.33 | 20.00 | 28.00 | 22.00 | 23.33 | 10.00 | 40.00 | 20.0 |
| | InternVL2.5 (Chen et al., 2024b) | 26B | 17.33 | 19.33 | 24.00 | 26.00 | 10.00 | 19.33 | 16.00 | 32.00 | 10.00 | 13.33 | 10.00 | 10.00 | 20.00 |
| | InternVL2.5 (Chen et al., 2024b) | 8B | 18.22 | 18.67 | 20.00 | 28.00 | 8.00 | 19.33 | 16.00 | 30.00 | 12.00 | 16.67 | 5.00 | 35.00 | 10.00 |
| | InternVL2.5 (Chen et al., 2024b) | 4B | 15.33 | 15.33 | 14.00 | 10.00 | 22.00 | 14.00 | 16.00 | 18.00 | 8.00 | 16.67 | 15.00 | 30.00 | 5.00 |
| | LLaVA Next (Li et al., 2024a) | 32B | 17.89 | 18.67 | 14.0 | 34.0 | 8.00 | 16.67 | 6.00 | 32.00 | 12.00 | 18.33 | 5.0 | 40.0 | 10.00 |
| | LLaVA Video (Zhang et al., 2024b) | 7B | 14.78 | 16.67 | 14.00 | 28.00 | 8.00 | 12.67 | 6.00 | 22.00 | 10.00 | 15.00 | 5.00 | 30.00 | 10.00 |
| | LLaVA OneVision (Li et al., 2024b) | 7B | 15.67 | 16.00 | 12.00 | 28.00 | 8.00 | 16.00 | 12.00 | 26.00 | 10.00 | 15.00 | 10.00 | 25.00 | 10.00 |
| | Qwen2.5 VL (Bai et al., 2025) | 32B | 16.22 | 20.00 | 6.00 | 36.00 | 18.00 | 15.33 | 4.00 | 24.00 | 18.00 | 13.33 | 0.00 | 30.00 | 10.00 |
| | Qwen2.5 VL (Bai et al., 2025) | 7B | 16.55 | 19.33 | 0.00 | 30.00 | 28.00 | 15.33 | 2.00 | 30.00 | 14.00 | 15.00 | 5.00 | 30.00 | 10.00 |
| Medium | GPT 5 (OpenAI, 2025) | - | 44.00 | 39.33 | 28.00 | 44.00 | 46.00 | 39.33 | 36.00 | 54.00 | 28.00 | 53.33 | 65.00 | 35.00 | 60.00 |
| | GPT 4o (Hurst et al., 2024) | - | 38.45 | 38.67 | 38.00 | 56.00 | 22.00 | 30.00 | 38.00 | 34.00 | 18.00 | 46.67 | 65.00 | 30.00 | 45.00 |
| | Gemini 2.5 pro (Google, 2025b) | - | 43.11 | 44.67 | 42.00 | 40.00 | 52.00 | 31.33 | 34.00 | 34.00 | 26.00 | 53.33 | 60.00 | 35.00 | 65.0 |
| | Gemini 2.5 flash think (Google, 2025a) | - | 39.78 | 39.33 | 32.00 | 38.00 | 48.00 | 30.00 | 34.00 | 28.00 | 28.00 | 50.00 | 65.00 | 15.00 | 70.00 |
| | Gemini 2.5 flash no-think (Google, 2025a) | - | **49.67** | 43.33 | 30.00 | 48.00 | 52.00 | 40.67 | 38.00 | 50.00 | 34.00 | 65.00 | 60.00 | 65.00 | 70.00 |
| | Gemini 1.5 pro (DeepMind, 2024) | - | 38.78 | 38.00 | 32.00 | 48.00 | 34.00 | 36.67 | 34.00 | 52.00 | 24.00 | 41.67 | 30.00 | 55.00 | 40.00 |
| | Claude 3.5 (Anthropic, 2024) | - | 39.67 | 38.00 | 26.00 | 40.00 | 48.00 | 36.00 | 32.00 | 54.00 | 22.00 | 45.00 | 50.00 | 35.00 | 50.0 |
| | InternVL2.5 (Chen et al., 2024b) | 26B | 28.67 | 31.33 | 28.00 | 58.00 | 8.00 | 24.67 | 12.00 | 50.00 | 12.00 | 30.00 | 25.00 | 45.00 | 20.00 |
| | InternVL2.5 (Chen et al., 2024b) | 8B | 34.33 | 30.00 | 20.00 | 58.00 | 12.00 | 34.67 | 32.00 | 50.00 | 22.00 | 38.33 | 40.00 | 45.00 | 30.00 |
| | InternVL2.5 (Chen et al., 2024b) | 4B | 32.22 | 29.33 | 28.00 | 44.00 | 16.00 | 34.00 | 30.00 | 54.00 | 18.00 | 33.33 | 35.00 | 40.00 | 25.00 |
| | LLaVA Next (Li et al., 2024a) | 32B | 26.11 | 24.67 | 18.0 | 40.0 | 16.00 | 25.33 | 18.0 | 40.0 | 18.00 | 28.33 | 25.0 | 40.0 | 20.00 |
| | LLaVA Video (Zhang et al., 2024b) | 7B | 24.00 | 25.33 | 24.00 | 36.00 | 16.00 | 21.00 | 16.00 | 26.00 | 18.00 | 26.67 | 15.00 | 45.00 | 20.00 |
| | LLaVA OneVision (Li et al., 2024b) | 7B | 23.67 | 23.33 | 20.00 | 34.00 | 16.00 | 22.67 | 20.00 | 32.00 | 16.00 | 25.00 | 20.00 | 35.00 | 20.00 |
| | Qwen2.5 VL (Bai et al., 2025) | 32B | 33.34 | 32.67 | 12.00 | 48.00 | 38.00 | 30.67 | 22.00 | 50.00 | 20.00 | 36.67 | 20.00 | 60.00 | 30.00 |
| | Qwen2.5 VL (Bai et al., 2025) | 7B | 28.00 | 24.67 | 16.00 | 24.00 | 34.00 | 26.00 | 24.00 | 26.00 | 28.00 | 33.33 | 35.00 | 20.00 | 45.00 |
| Easy | GPT 5 (OpenAI, 2025) | - | 52.00 | 47.33 | 42.00 | 42.00 | 58.00 | 48.67 | 46.00 | 46.00 | 54.00 | 60.00 | 65.00 | 35.00 | 80.00 |
| | GPT 4o (Hurst et al., 2024) | - | 40.67 | 35.33 | 30.00 | 28.00 | 48.00 | 36.67 | 24.00 | 38.00 | 48.00 | 50.00 | 45.00 | 50.00 | 55.00 |
| | Gemini 2.5 pro (Google, 2025b) | - | **52.56** | 56.00 | 60.00 | 48.00 | 60.00 | 40.00 | 40.00 | 36.00 | 44.00 | 61.67 | 75.00 | 35.00 | 75.0 |
| | Gemini 2.5 flash think (Google, 2025a) | - | 50.67 | 49.33 | 40.00 | 46.00 | 62.00 | 46.00 | 46.00 | 44.00 | 48.00 | 56.67 | 55.00 | 40.00 | 75.00 |
| | Gemini 2.5 flash no-think (Google, 2025a) | - | 50.78 | 49.33 | 36.00 | 52.00 | 60.00 | 48.00 | 40.00 | 50.00 | 54.00 | 55.00 | 60.00 | 50.00 | 55.00 |
| | Gemini 1.5 pro (DeepMind, 2024) | - | 43.00 | 45.33 | 36.00 | 44.00 | 56.00 | 42.00 | 48.00 | 32.00 | 46.00 | 41.67 | 35.00 | 50.00 | 40.00 |
| | Claude 3.5 (Anthropic, 2024) | - | 42.45 | 38.00 | 34.00 | 38.00 | 42.00 | 42.00 | 30.00 | 56.00 | 42.00 | 46.67 | 40.00 | 45.00 | 55.0 |
| | InternVL2.5 (Chen et al., 2024b) | 26B | 36.11 | 35.33 | 36.00 | 44.00 | 26.00 | 34.67 | 28.00 | 46.00 | 30.00 | 38.33 | 30.00 | 40.00 | 45.00 |
| | InternVL2.5 (Chen et al., 2024b) | 8B | 38.44 | 36.67 | 28.00 | 46.00 | 36.00 | 35.33 | 32.00 | 42.00 | 32.00 | 43.33 | 60.00 | 40.00 | 30.00 |
| | InternVL2.5 (Chen et al., 2024b) | 4B | 40.33 | 43.33 | 42.00 | 50.00 | 38.00 | 39.33 | 30.00 | 44.00 | 44.00 | 38.33 | 35.00 | 60.00 | 20.00 |
| | LLaVA Next (Li et al., 2024a) | 32B | 33.22 | 36.67 | 36.00 | 42.00 | 32.00 | 31.33 | 36.00 | 32.00 | 26.00 | 31.67 | 35.00 | 30.00 | 30.00 |
| | LLaVA Video (Zhang et al., 2024b) | 7B | 33.22 | 33.33 | 34.00 | 38.00 | 28.00 | 34.67 | 34.00 | 38.00 | 32.00 | 31.67 | 35.00 | 30.00 | 30.00 |
| | LLaVA OneVision (Li et al., 2024b) | 7B | 33.22 | 33.33 | 34.00 | 38.00 | 28.00 | 34.67 | 34.00 | 38.00 | 32.00 | 31.67 | 35.00 | 30.00 | 30.00 |
| | Qwen2.5 VL (Bai et al., 2025) | 32B | 52.45 | 50.00 | 34.00 | 56.00 | 60.00 | 50.67 | 40.00 | 54.00 | 58.00 | 56.67 | 55.00 | 60.00 | 55.00 |
| | Qwen2.5 VL (Bai et al., 2025) | 7B | 39.89 | 33.33 | 28.00 | 18.00 | 54.00 | 38.00 | 48.00 | 16.00 | 50.00 | 48.33 | 55.00 | 30.00 | 60.00 |

Figure 7: A question and answer example: For each each scenario reasoning setting, we include three types of video lengths: short, medium, and long. Each video length includes tasks designed to evaluate temporal reasoning, spatial reasoning, and intent reasoning.

dynamic scenarios that test rapid perception and reasoning. **(b) Video Categories:** The benchmark spans three safety-critical domains. Vehicle Accident, which primarily involves traffic and safety-related scenarios, comprises 83.0% of the videos. airplane navigation accounts for 10.2%, and ship motion makes up 6.8%. This distribution highlights both the practical importance of land-based reasoning and the inclusion of underrepresented domains such as maritime and aviation environments. **(c) Understanding and Reasoning Styles:** AccidentBench supports three major understanding and reasoning types, with a relatively balanced distribution: *spatial reasoning* (35.4%), *temporal*

Table 10: Evaluation of AccidentBench in the **Ship Motion** using **River** and **Ocean** Videos, categorized by reasoning types, based on a subset of the dataset.

| Difficulty | Models | Size | Over. Avg. | River Scenarios Avg. | Temporal | Spatial | Intent | Ocean Scenarios Avg. | Temporal | Spatial | Intent |
|---|---|---|---|---|---|---|---|---|---|---|---|
| Hard | GPT 5 (OpenAI, 2025) | - | **38.36** | 48.72 | 46.15 | 30.77 | 69.23 | 28.00 | 30.00 | 32.00 | 22.00 |
| | GPT4o (Hurst et al., 2024) | - | 22.10 | 28.20 | 38.46 | 26.92 | 19.23 | 16.00 | 18.00 | 18.00 | 12.00 |
| | Gemini 2.5 pro (Google, 2025b) | - | 29.64 | 34.62 | 23.08 | 34.62 | 46.15 | 24.67 | 38.00 | 16.00 | 20.00 |
| | Gemini 2.5 flash think (Google, 2025a) | - | 27.36 | 32.05 | 30.77 | 26.92 | 38.46 | 22.67 | 30.00 | 22.00 | 16.00 |
| | Gemini 2.5 flash no-think (Google, 2025a) | - | 27.44 | 28.21 | 42.31 | 19.23 | 23.08 | 26.67 | 36.00 | 20.00 | 24.00 |
| | Gemini 1.5 pro (DeepMind, 2024) | - | 26.02 | 26.92 | 23.08 | 30.77 | 26.92 | 25.11 | 34.00 | 20.93 | 20.41 |
| | Claude 3.5 (Anthropic, 2024) | - | 25.44 | 28.20 | 19.23 | 19.23 | 46.15 | 22.67 | 26.00 | 22.00 | 20.00 |
| | InternVL2.5 (Chen et al., 2024b) | 26B | 22.54 | 23.08 | 15.38 | 19.23 | 34.62 | 22.00 | 18.00 | 28.00 | 20.00 |
| | InternVL2.5 (Chen et al., 2024b) | 8B | 21.90 | 21.79 | 7.69 | 26.92 | 30.77 | 22.00 | 16.00 | 28.00 | 22.00 |
| | InternVL2.5 (Chen et al., 2024b) | 4B | 20.92 | 20.51 | 19.23 | 19.23 | 23.08 | 21.33 | 16.00 | 26.00 | 22.00 |
| | LLaVA Next (Li et al., 2024a) | 32B | 14.39 | 11.54 | 7.69 | 19.23 | 7.69 | 15.33 | 8.00 | 30.00 | 8.00 |
| | LLaVA Video (Zhang et al., 2024b) | 7B | 14.00 | 16.67 | 15.38 | 23.08 | 11.54 | 11.33 | 8.00 | 20.00 | 6.00 |
| | LLaVA OneVision (Li et al., 2024b) | 7B | 15.67 | 16.67 | 11.54 | 26.92 | 11.54 | 14.67 | 8.00 | 28.00 | 8.00 |
| | Qwen2.5 VL (Bai et al., 2025) | 32B | 13.39 | 14.10 | 7.69 | 23.08 | 11.54 | 12.67 | 8.0 | 24.0 | 6.00 |
| | Qwen2.5 VL (Bai et al., 2025) | 7B | 14.67 | 16.67 | 7.69 | 30.77 | 11.54 | 12.67 | 6.00 | 24.00 | 8.00 |
| Medium | GPT 5 (OpenAI, 2025) | - | **51.80** | 60.26 | 53.85 | 46.15 | 80.77 | 43.33 | 56.00 | 48.00 | 26.00 |
| | GPT 4o (Hurst et al., 2024) | - | 38.49 | 42.31 | 50.00 | 53.85 | 23.08 | 34.67 | 36.00 | 48.00 | 20.00 |
| | Gemini 2.5 pro (Google, 2025b) | - | 41.77 | 44.87 | 30.77 | 61.54 | 42.31 | 38.67 | 48.00 | 46.00 | 22.00 |
| | Gemini 2.5 flash think (Google, 2025a) | - | 48.26 | 53.85 | 61.54 | 57.70 | 42.31 | 42.67 | 52.00 | 42.00 | 34.00 |
| | Gemini 2.5 flash no-think (Google, 2025a) | - | 46.12 | 50.00 | 46.15 | 57.69 | 46.15 | 42.00 | 56.00 | 44.00 | 26.00 |
| | Gemini 1.5 pro (DeepMind, 2024) | - | 46.31 | 53.84 | 46.15 | 65.38 | 50.00 | 38.78 | 34.00 | 49.02 | 33.33 |
| | Claude 3.5 (Anthropic, 2024) | - | 38.62 | 35.90 | 34.62 | 50.00 | 23.08 | 41.33 | 42.00 | 54.00 | 28.00 |
| | InternVL2.5 (Chen et al., 2024b) | 26B | 41.77 | 44.87 | 30.77 | 57.69 | 46.15 | 38.67 | 24.00 | 62.00 | 30.00 |
| | InternVL2.5 (Chen et al., 2024b) | 8B | 41.08 | 46.15 | 34.62 | 61.54 | 42.31 | 36.00 | 34.00 | 60.00 | 14.00 |
| | InternVL2.5 (Chen et al., 2024b) | 4B | 44.36 | 48.72 | 23.08 | 65.38 | 57.69 | 40.00 | 28.00 | 60.00 | 32.00 |
| | LLaVA Next (Li et al., 2024a) | 32B | 20.88 | 23.08 | 11.54 | 38.46 | 19.23 | 18.67 | 10.00 | 30.00 | 16.00 |
| | LLaVA Video (Zhang et al., 2024b) | 7B | 21.92 | 20.51 | 19.23 | 26.92 | 15.38 | 23.33 | 20.00 | 30.00 | 20.00 |
| | LLaVA OneVision (Li et al., 2024b) | 7B | 22.54 | 23.08 | 19.23 | 30.77 | 19.23 | 22.00 | 14.00 | 34.00 | 18.00 |
| | Qwen2.5 VL (Bai et al., 2025) | 32B | 33.31 | 34.62 | 19.23 | 50.00 | 34.62 | 32.00 | | 50.00 | 26.00 |
| | Qwen2.5 VL (Bai et al., 2025) | 7B | 24.08 | 29.49 | 19.23 | 30.77 | 38.46 | 18.67 | 18.00 | 26.00 | 12.00 |
| Easy | GPT 5 (OpenAI, 2025) | - | **63.00** | 66.67 | 61.54 | 50.00 | 88.46 | 59.33 | 78.00 | 48.00 | 52.00 |
| | GPT 4o (Hurst et al., 2024) | - | 50.51 | 57.69 | 57.69 | 50.00 | 65.38 | 43.33 | 66.00 | 34.00 | 30.00 |
| | Gemini 2.5 pro (Google, 2025b) | - | 61.05 | 64.10 | 57.69 | 57.69 | 76.92 | 58.00 | 72.00 | 50.00 | 52.00 |
| | Gemini 2.5 flash think (Google, 2025a) | - | 62.03 | 65.39 | 80.77 | 42.31 | 73.08 | 58.67 | 70.00 | 52.00 | 54.00 |
| | Gemini 2.5 flash no-think (Google, 2025a) | - | 58.18 | 57.69 | 57.69 | 38.46 | 76.92 | 58.67 | 80.00 | 42.00 | 54.00 |
| | Gemini 1.5 pro (DeepMind, 2024) | - | 50.69 | 52.56 | 42.31 | 61.54 | 53.85 | 48.81 | 50.00 | 46.43 | 50.00 |
| | Claude 3.5 (Anthropic, 2024) | - | 49.39 | 47.44 | 50.00 | 53.85 | 38.46 | 51.33 | 62.00 | 52.00 | 40.00 |
| | InternVL2.5 (Chen et al., 2024b) | 26B | 55.05 | 64.10 | 65.38 | 57.69 | 69.23 | 46.00 | 50.00 | 50.00 | 38.00 |
| | InternVL2.5 (Chen et al., 2024b) | 8B | 53.47 | 60.26 | 69.23 | 46.15 | 65.38 | 46.67 | 46.00 | 54.00 | 40.00 |
| | InternVL2.5 (Chen et al., 2024b) | 4B | 53.87 | 56.41 | 53.85 | 57.69 | 57.69 | 51.33 | 52.00 | 56.00 | 46.00 |
| | LLaVA Next (Li et al., 2024a) | 32B | 35.59 | 37.18 | 26.92 | 53.85 | 30.77 | 34.00 | 30.00 | 38.00 | 34.00 |
| | LLaVA Video (Zhang et al., 2024b) | 7B | 31.03 | 32.05 | 30.77 | 34.62 | 30.77 | 30.00 | 22.00 | 38.00 | 30.00 |
| | LLaVA OneVision (Li et al., 2024b) | 7B | 33.00 | 33.33 | 34.62 | 34.62 | 19.23 | 32.67 | 28.00 | 38.00 | 32.00 |
| | Qwen2.5 VL (Bai et al., 2025) | 32B | 52.77 | 61.54 | 53.85 | 61.54 | 69.23 | 44.00 | 40.00 | 54.00 | 38.00 |
| | Qwen2.5 VL (Bai et al., 2025) | 7B | 31.31 | 34.62 | 38.46 | 19.23 | 46.15 | 28.00 | 36.00 | 22.00 | 26.00 |

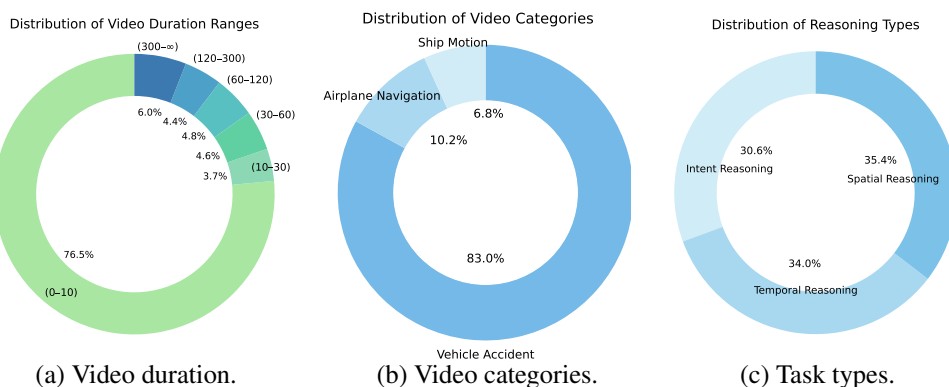

(a) Video duration.  (b) Video categories.  (c) Task types.

Figure 8: Distribution of video and task properties in the AccidentBench benchmark.

*reasoning* (34.0%), and *intent reasoning* (30.6%). This design ensures comprehensive evaluation across key dimensions essential for real-world multimodal understanding. Overall, the dataset provides a rich and diverse collection of real-world video scenarios across multiple modalities and time scales, offering a rigorous benchmark for evaluating multimodal understanding and reasoning in safety-critical environments.

## G  Detailed Benchmark Comparisons of Reasoning Requirements

Table 2 compares AccidentBench with existing benchmarks in terms of scale and characteristics. Here, we provide a more detailed discussion of the differences in reasoning requirements emphasized by these benchmarks. In particular, we focus on how AccidentBench extends reasoning challenges beyond what existing benchmarks typically evaluate.

Specifically, AccidentBench focuses on safety-critical traffic scenarios, introducing tasks that require deeper intent reasoning under high-stakes conditions. By contrast, existing safety benchmarks such as DriveLM (Sima et al., 2024b) and DriveBench (Xie et al., 2025) primarily focus on routine driving scenes and basic perception or reasoning tasks. AccidentBench is designed to cover high-stakes accident scenarios, which are crucial for evaluating autonomous systems under failure-prone or edge-case conditions.

To achieve this, AccidentBench incorporates reasoning tasks that emphasize:

- **Multi-agent reasoning:** understanding interactions between multiple agents, including vehicles, pedestrians, and cyclists.

- **Multi-event reasoning:** interpreting sequences of interrelated events rather than isolated single-frame observations.

- **Temporal-causal reasoning:** predicting potential outcomes or counterfactual alternatives based on observed dynamics and causal dependencies.

This design better reflects real-world decision-making complexity, requiring models to reason about spatial dynamics, causal chains, and future intent, as opposed to static or one-shot judgments commonly encountered in prior benchmarks. By providing a higher level of reasoning granularity and realism in safety-critical contexts, AccidentBench complements existing driving benchmarks and enables more rigorous evaluation of multimodal spatial and temporal intelligence.

## H  Annotation Pipeline and Intent Reasoning Examples

### H.1  Annotation Pipeline

Our data collection and annotation process was designed to ensure both safety-critical relevance and scene diversity while minimizing potential biases.

Data Collection and Selection: We collected over 2,000 videos and curated approximately 2,000 videos from public platforms (e.g., YouTube) and existing open datasets, focusing on safety-critical accident scenarios. The selection prioritized clips that:(1) Clearly depict safety-critical scenarios (e.g., vehicle–pedestrian, vehicle–vehicle, vehicle–environment). (2) Cover diverse contexts, including variations in weather, lighting, road types, and viewpoints. (3) Provide sufficient temporal length to capture cause–and–effect dynamics.

Annotation Pipeline: (1) Initial Filtering: Videos were manually screened to remove low-quality or ambiguous footage. (2) Task Definition: After pilot discussions, we defined key annotation tasks (e.g., temporal understanding and reasoning, spatial relationships, causal factors). (3) Assignment: Annotators independently labeled their assigned subsets following standardized guidelines, including specifications on question types, option formats, and contextual consistency. (4) Cross-Review and Refinement: Annotations were peer-reviewed by annotators, followed by reconciliation discussions to resolve inconsistencies. (5) Final Validation: performed quality checks to ensure consistency, completeness, and neutrality.

## H.2 INTENT REASONING EXAMPLES

**Intent reasoning example 1**:

**Question**:
If you are the colliding and violating the rules car starting from its initial position, and you want to navigate in the same direction as the stationary white car's intention direction, you will perform the following actions (Note: for each [please fill in]): {1. [please fill in] 2. [please fill in] 3. [please fill in]. You will find an intersection and find a way to go to that direction.

**Choices**: A. go back, B. turn left, C. turn right, D. turn on the left turn signal, E. turn on the right turn signal, F. start the engine, G. Go forward

**Ground Truth**: B

**Options**:

- A. CED

- B. DBG

- C. CGD

- D. CEF

- E. ECD

- F. FBD

- G. DCA

- H. GEB

- I. EFB

- J. EDB

- K. CDF

- L. DCG

**Intent reasoning example 2**:

**Question**:
If you are the red car's driver starting from its initial position, and you want to navigate toward the leftmost lane in the direction that the collided sedan intends to travel, you will perform the following actions (Note: for each step, please fill in): {1. go forward 2. [please fill in]} 3. [please fill in] 4. [please fill in].

**Choices**: A. go back, B. turn left, C. turn right, D. turn on the left turn signal, E. turn on the right turn signal, F. approaching the next U-turn intersection, G. pass the current intersection, H. observe the traffic situation, I. stop and go back, J. speed up and turn left, K. find a suitable traffic situation

**Ground Truth**: H

**Options**:

- A. JHFB

- B. FKDH

- C. CEHK

- D. DCEH

- E. BFDK

- F. BHEK

- G. GHJK

- H. DHKB

- I. KDCB

- J. HKJD

- K. HJKB

- L. CKBE

**Intent reasoning example 3**:

**Question**:
What is the entry order of the first accident vehicle into the scene?

**Ground Truth**: B

**Options**:

- A. 1

- B. 2

- C. 4

- D. 6

**Intent reasoning example 4**:

**Question**:
How many moving vehicles have been or are going to be passed by the ego vehicle, and those vehicles are traveling in the opposite direction of the ego vehicle's initial navigation?

**Ground Truth**: C

**Options**:

- E. 9

- G. 4

- H. 6

- L. 0

- K. 10

- J. 11

- B. 1

- I. 8

- A. 7

- C. 5

- F. 2

- D. 3

**Intent reasoning example 5**:

**Question**:
If you are the driver of the accident car and you want to avoid the collision, then you will perform the following action when passing by the bicycle.

**Ground Truth**: A

**Options**:

- A. reduce speed

- B. speed up

- C. turn right

- D. turn left

## I  DETAILED EXPERIMENT SETTINGS

The datasets are used solely for academic research. They are employed only to evaluate model performance and are not used for model training. In our experiments, we build upon the `lmms-eval` framework (Zhang et al., 2024a) as the foundation for our benchmark and extend it to support the

specific requirements of AccidentBench. All experiments with open-source models were conducted on a Linux system equipped with $8 \times$ NVIDIA A100 GPUs, and experiments with closed-source models were run on a single NVIDIA A100 GPU. Key hyperparameters used for model evaluation are summarized in Table 11.

Table 11: Key parameters used in the experiments.

| Parameters | value | Parameters | value |
|---|---|---|---|
| sample size | 1 | number of processes | 8 |
| max pixels (Qwen 2.5) | 12845056 | use-flash-attention-2 (Qwen 2.5) | False |
| interleave visuals (Qwen 2.5) | True | enable-chunked-prefill (InternVL 2.5) | True |
| gpu-memory-utilization (InternVL 2.5) | 0.6 | max-num-seqs (InternVL 2.5) | 1 |
| conv-template (LLava-Video) | qwen-1-5 | video-decode-backend (LLava-Video) | record |
| max-frames-num (LLava-Video) | 22 | mm-spatial-pool-mode (LLava-Video) | average |
| mm-newline-position (LLava-Video) | grid | mm-resampler-location (LLava-Video) | after |
| conv-template (llava-onevision) | qwen-1-5 | device-map (llava-onevision) | auto |
| model-name (llava-onevision) | llava-qwen | | |

