# OpenReview forum: "AccidentBench: Benchmarking Multimodal Understanding and Reasoning in Vehicle Accidents and Beyond"
_ICLR.cc/2026/Conference — Submitted to ICLR 2026_

### Official Review · Reviewer_kyon · 2025-10-27

**Soundness:** 3
**Presentation:** 3
**Contribution:** 3
**Rating:** 4
**Confidence:** 5

**Summary:**

This paper introduces AccidentBench, a large-scale video benchmark designed to rigorously evaluate multimodal models’ understanding and reasoning capabilities in safety-critical, dynamic real-world environments.

**Strengths:**

1. The benchmark presented in this paper is well-designed. It explicitly decomposes evaluation into three key reasoning dimensions: temporal, spatial, and intentional reasoning. This systematic framework is crucial for conducting a fine-grained analysis of model capabilities.
2 The dataset comprises over 19,000 human-annotated question-answer pairs. Such rigorous annotation standards are essential for ensuring the reliability and precision required in complex reasoning tasks.
3. The experiments are relatively comprehensive. The authors evaluate a wide range of state-of-the-art multimodal models, including both closed-source and open-source models.

**Weaknesses:**

1. There are discrepancies in the reported percentages for the distribution across the three domains in multiple sections of the paper. The Introduction states the distribution as 83.0% for Vehicle accident scenarios, 10.2% for airplane navigation scenarios, and 6.8% for ship motion scenarios. However, Section 3.1 reports these percentages as 83%, 10.8%, and 6.2%, respectively.
2. 2000 videos are insufficient to represent the diversity of real-world accident scenarios adequately. The authors may further expand the dataset's scale.
3. The dataset’s duration distribution is heavily skewed toward short clips (76.5% under 10 seconds), which raises concerns about extrapolation validity regarding the paper’s key conclusion that model performance drops significantly on long videos. It is recommended to increase the proportion of long videos or to provide a dedicated, more comprehensive evaluation on a long-video subset to substantiate this finding.

**Questions:**

1. The authors primarily focus on accident videos from surveillance scenes. Do the authors plan to include and annotate videos captured from a driver’s viewpoint in future work? (e.g., dashcam videos)
2. The paper mentions that the videos were collected from public platforms such as YouTube and from existing datasets. While this is common practice, it would be beneficial to include (perhaps in the appendix) more details about the data collection and filtering process. For example, how are the 2,000 videos selected from a vast pool of publicly available data? Please give an annotation pipeline. What specific criteria are used to ensure scene diversity and to avoid potential dataset biases?
3. Apart from the number of answer options, is there a computable difficulty categorization based on video attributes such as occlusion, viewpoint changes, number of objects, or motion dynamics?

**Details Of Ethics Concerns:**

NO concerns.

---

> ### Author Response · Authors · 2025-11-21
>
> Dear Reviewer kyon,
>
> We thank the reviewer for the positive and thoughtful evaluation. We appreciate your recognition of our benchmark’s structured decomposition into temporal, spatial, and intent reasoning, as well as the rigor of our 19,000 human-annotated QA pairs. We address your comments in detail below.
>
> > **Q1:** There are discrepancies in the reported percentages for the distribution across the three domains in multiple sections of the paper. The Introduction states the distribution as 83.0% for Vehicle accident scenarios, 10.2% for airplane navigation scenarios, and 6.8% for ship motion scenarios. However, Section 3.1 reports these percentages as 83%, 10.8%, and 6.2%, respectively.
>
> **A1:** Thanks for your careful reviews. This is a typo, it should be 83%, 10.2%, and 6.8%, respectively. We applogy to this typo confuse you. We have fixed this typo.
>
> > **Q2:** 2000 videos are insufficient to represent the diversity of real-world accident scenarios adequately. The authors may further expand the dataset's scale.
>
> **A2:** We appreciate the reviewer’s insightful comment. Our dataset is designed to balance diversity, evaluation efficiency, and usability. While a much larger dataset could further increase coverage, it would also introduce substantial evaluation overhead, making systematic benchmarking more difficult. Our current scale of ~2,000 videos provides a practical and comprehensive testbed that captures diverse accident scenarios while remaining manageable for reproducible evaluation across models.
>
> Moreover, as illustrated in **Figure 2** and **Table 1**, our dataset captures **diverse conditions across road types, weather patterns, and participant interactions**, ensuring broad coverage of real-world accident contexts. Furthermore, as shown in **Table 2**, our dataset size is **comparable to or larger than several recent state-of-the-art benchmarks**, such as **TempCompass** (410 videos, Liu et al., ACL 2024), **VSI-Bench** (288 videos, Yang et al., CVPR 2025), **Video-MMMU** (300 videos, Hu et al., 2025), and **Video-MMLU** (1,065 videos, Song et al., 2025).
>
>
> > **Q3:** The dataset’s duration distribution is heavily skewed toward short clips (76.5% under 10 seconds), which raises concerns about extrapolation validity regarding the paper’s key conclusion that model performance drops significantly on long videos. It is recommended to increase the proportion of long videos or to provide a dedicated, more comprehensive evaluation on a long-video subset to substantiate this finding.
>
> **A3:** We appreciate the reviewer’s thoughtful comment. Our dataset provides rich coverage with 19,000 QA tasks and 2,000 videos. Although the full dataset contains many short clips, our evaluation protocol uses a balanced sampling strategy to ensure that models are tested on a representative mix of short, medium, and long videos. This approach commonly adopted in other benchmarks, allows efficient yet fair evaluation across duration categories. As shown in our evaluation split (3,798 QA pairs), short videos's qa account for 49.62% (1,884 out of 3,798), demonstrating that the test sets used for model comparison are not dominated by short clips. We also conducted ablation studies to confirm that this sampling strategy is appropriate and does not affect our key findings.
>
> > **Q4:** The authors primarily focus on accident videos from surveillance scenes. Do the authors plan to include and annotate videos captured from a driver’s viewpoint in future work? (e.g., dashcam videos)
>
> **A4:** Thank you for the valuable suggestion. Our dataset already includes a portion of videos captured from a driver’s viewpoint, such as dashcam and in-vehicle recordings, in addition to fixed surveillance footage. These first-person perspectives enhance the dataset’s diversity and support the study of egocentric spatial reasoning and driver-intent understanding.

---

> > ### Author Response · Authors · 2025-11-21
> >
> > > **Q5:** The paper mentions that the videos were collected from public platforms such as YouTube and from existing datasets. While this is common practice, it would be beneficial to include (perhaps in the appendix) more details about the data collection and filtering process. For example, how are the 2,000 videos selected from a vast pool of publicly available data? Please give an annotation pipeline. What specific criteria are used to ensure scene diversity and to avoid potential dataset biases?
> >
> > **A5:** Thank you for the helpful suggestion. Our data collection and annotation process was designed to ensure both safety-critical relevance and scene diversity while minimizing potential biases, we have updated it in our revision version's Appendix H.1 (also see here: https://anonymous.4open.science/r/m4r-436E/data_results/annotation_pipeline.png).
> >
> > Data Collection and Selection: We collected over 2,000 videos and curated approximately 2,000 videos from public platforms (e.g., YouTube) and existing open datasets, focusing on safety-critical accident scenarios. The selection prioritized clips that:(1) Clearly depict safety-critical scenarios (e.g., vehicle–pedestrian, vehicle–vehicle, vehicle–environment). (2) Cover diverse contexts, including variations in weather, lighting, road types, and viewpoints. (3) Provide sufficient temporal length to capture cause–and–effect dynamics.
> >
> > Annotation Pipeline: (1) Initial Filtering: Videos were manually screened to remove low-quality or ambiguous footage. (2) Task Definition: After pilot discussions, we defined key annotation tasks (e.g., temporal understanding and reasoning, spatial relationships, causal factors). (3) Assignment: Annotators independently labeled their assigned subsets following standardized guidelines, including specifications on question types, option formats, and contextual consistency. (4) Cross-Review and Refinement: Annotations were peer-reviewed by annotators, followed by reconciliation discussions to resolve inconsistencies. (5) Final Validation: performed quality checks to ensure consistency, completeness, and neutrality.
> >
> > > **Q6:** Apart from the number of answer options, is there a computable difficulty categorization based on video attributes such as occlusion, viewpoint changes, number of objects, or motion dynamics?
> >
> > **A6:** Thank you for the insightful question.  We agree that establishing a computable difficulty categorization would provide valuable insights.  Our dataset indeed includes videos exhibiting diverse difficulty factors, such as viewpoint shifts, occlusions, varying object counts, complex motion dynamics, and multi-agent interactions. We evaluate model performance in these diverse scenarios under the same settings. Moreover, we conducted new experiments and discuss the geographic and weather accuracy in our revised paper, also see here for the results https://anonymous.4open.science/r/m4r-436E/data_results/geo_weather_accuracy.png, we observe that model performance in adverse conditions—such as snow, rain, and nighttime lighting, as well as in urban environments is worse than in favorable weather (sunny, bright) and rural road settings.

---

> > ### Comment · Reviewer_kyon · 2025-11-26
> >
> > Based on the rebuttal, the authors do not have positive and strong evidence to address my concerns. Therefore, I will keep my rating in this round.

---

> > > ### Author Response · Authors · 2025-11-26
> > >
> > > Dear Reviewer kyon
> > >
> > > Thank you for taking the time to evaluate our rebuttal. We fully respect your decision; however, we would appreciate it if you could kindly indicate which specific points you believe we did not adequately address. We have made every effort to provide detailed clarifications, additional analyses, and revisions in direct response to the concerns raised.
> > >
> > > If there are particular aspects where our explanations were unclear or insufficient, we would greatly value the opportunity to provide further evidence, analysis, or revisions. Our goal is to ensure that all concerns are fully resolved and that the contributions of this work are assessed as accurately as possible.
> > >
> > > We sincerely thank you again for your thoughtful review and consideration.
> > >
> > > Warm regards,
> > >
> > > The Authors

---

> > > > ### Comment · Reviewer_kyon · 2025-11-27
> > > >
> > > > Thanks for the author's careful rebuttal. The main concerns in my initial comments are:
> > > > 1. 2000 videos are insufficient to represent the diversity of real-world accident scenarios adequately. According to the authors' rebuttal, the data scale issue is currently limited. In previous works, such as Roadsocial [1] and Echotraffic [2], many attempts have been made to explore the problem of understanding multimodal traffic accidents using over 10k sequences. Therefore, the data scale in this work is limited for exploring further new insights.
> > > > 2. The comparison methods in this work do not contain the identical works in 2025, such as [1-2], where only two fine-tuned versions of Qwen2.5VL are involved. In addition, many related accident video understanding works are absent from the Table. 2, which cannot reflect the newest update.
> > > >
> > > > [1] RoadSocial: A Diverse VideoQA Dataset and Benchmark for Road Event Understanding from Social Video Narratives, CVPR, 2025.
> > > > [2] Echotraffic: Enhancing traffic anomaly understanding with audio-visual insights, in CVPR 2025.

---

> ### Author Response · Authors · 2025-11-27
>
> Dear Reviewer Kyon,
>
> Thank you for your valuable and prompt feedback, as well as your engagement during the rebuttal process. Below is our response to your comments.
>
> > **Q1:** 2000 videos are insufficient to represent the diversity of real-world accident scenarios adequately. According to the authors' rebuttal, the data scale issue is currently limited. In previous works, such as Roadsocial [1] and Echotraffic [2], many attempts have been made to explore the problem of understanding multimodal traffic accidents using over 10k sequences. Therefore, the data scale in this work is limited for exploring further new insights. [1] RoadSocial: A Diverse VideoQA Dataset and Benchmark for Road Event Understanding from Social Video Narratives, CVPR, 2025.
> [2] Echotraffic: Enhancing traffic anomaly understanding with audio-visual insights, in CVPR 2025.
>
> **A1:** Thank you for raising this important point, we have revised our paper and introduced the related work [1] and [2] in our paper (Also see here: https://anonymous.4open.science/r/m4r-436E/data_results/related-work-add.png). While we agree that larger datasets such as RoadSocial [1] and EchoTraffic [2] provide valuable contributions, their goals, scope, and task formulations differ substantially from ours. RoadSocial focuses on narration-driven road-event understanding, and EchoTraffic emphasizes audio-visual anomaly interpretation with audio-based cues and rich textual explanations. For example, EchoTraffic contains 29,865 videos with audio-aligned annotations designed for five anomaly-centric tasks such as anomaly timing, response, and prevention.
>
> In contrast, AccidentBench is deliberately designed as a fine-grained multimodal reasoning benchmark, not a large-scale anomaly detection corpus. Our focus is on video-based multiple-choice reasoning—temporal, spatial, and intent reasoning—across three domains (land, air, water), emphasizing causal modeling, multi-agent intent prediction, and safety-critical decision scenarios, which are not covered by RoadSocial or EchoTraffic. Although our dataset scale is about ~2,000 videos, 19,000 QA pairs, it is comparable to or larger than recent reasoning-type benchmarks (e.g., TempCompass (410 videos, Liu et al., ACL 2024), VSI-Bench (288 videos, Yang et al., CVPR 2025), Video-MMMU (300 videos, Hu et al., 2025), and Video-MMLU (1,065 videos, Song et al., 2025).) and is sufficient for the type of controlled, diagnostic evaluation we target. Expanding the dataset further is part of our planned future work, but the current scale already enables high-resolution reasoning assessment, which differs fundamentally from the objectives of RoadSocial and EchoTraffic.
>
> Importantly, the benchmark’s design allows it to reveal systematic reasoning performance in state-of-the-art models, for example, ChatGPT 5 accuracy dropping to ~18% on long-video tasks, highlighting its effectiveness as a safety-critical evaluation tool.
>
>
>
> > **Q2:** The comparison methods in this work do not contain the identical works in 2025, such as [1-2], where only two fine-tuned versions of Qwen2.5VL are involved. In addition, many related accident video understanding works are absent from the Table. 2, which cannot reflect the newest update.
>
>
> **A2:** Thank you for your valuable comments. We agree that recent works such as RoadSocial [1] and EchoTraffic [2] provide valuable contributions to traffic-event understanding. However, these datasets address different problem settings, narration-driven road-event QA and audio-visual anomaly analysis, respectively, whereas AccidentBench focuses on multiple-choice temporal, spatial, and intent reasoning across land, air, and water domains. Consequently, their evaluation pipelines are not directly compatible with our Multiple-Choice Question–based reasoning tasks, and their publicly released models are not designed for the fine-grained diagnostic reasoning that AccidentBench evaluates.
>
> Moreover, for our comparisons, we selected methods that natively support video-based multiple-choice reasoning or can be reliably adapted to this format, ensuring fair and consistent evaluation, such as ChatGPT 5, ChatGPT 4o, Gemini 2.5 Pro, LLaVA series, Qwen2.5 VL, and InternVL2.5 models. We included both proprietary and open-source multimodal LLMs to represent the current state of the field. We have revised our paper and now incorporated more new works into Table 2 (also available here for convenience: https://anonymous.4open.science/r/m4r-436E/data_results/table-2-updated.png).

---

### Official Review · Reviewer_m4t8 · 2025-10-31

**Soundness:** 3
**Presentation:** 3
**Contribution:** 2
**Rating:** 4
**Confidence:** 3

**Summary:**

AccidentBench introduces a large-scale benchmark for evaluating multimodal large language models on traffic accident understanding, focusing on reasoning, prediction, and prevention. The dataset contains a diverse collection of real-world accident scenarios, annotated with causal, temporal, and spatial relations to test models' comprehension beyond visual recognition. Experimental results across existing MLLMs reveal significant performance gaps in causal reasoning and intervention prediction, underscoring the need for better scene-grounded understanding.

**Strengths:**

1. The dataset has rich multimodal annotations that combine spatial, temporal, and causal information, offering fine-grained supervision for complex event understanding.
2. Evaluates a broad range of models and provides detailed failure analysis, highlighting specific reasoning challenges in current systems.

**Weaknesses:**

1. The dataset is relatively narrowly focused. It focuses on traffic accidents, which may restrict applicability to broader event reasoning domains.
2. The idea of developing vision reasoning datasets based on accidents is not new and has been explored in previous papers such as TrafficQA, which can weaken the novelty of the dataset and the authors didn't compare the differences.
3. A minor point is that it would be great to also collect a small amount of in-domain training data and show experimental results.

**Questions:**

What are the differences between your dataset and existing works such as TrafficQA?

---

> ### Author Response · Authors · 2025-11-21
>
> Dear Reviewer m4t8,
>
>
> We thank the reviewer for the constructive feedback and thoughtful assessment. We appreciate your recognition of the benchmark’s rich multimodal annotations and its ability to reveal concrete reasoning limitations through detailed model evaluations and failure analyses. We address your concerns below.
>
> > **Q1:** The dataset is relatively narrowly focused. It focuses on traffic accidents, which may restrict applicability to broader event reasoning domains.
>
> **A1:** Traffic accident understanding and reasoning is the key and bootleneck for AI in autonomous driving, if the model can understand and reason well in accident scenarios, that could very useful for ai models in real-world applications. Moreover, While our dataset centers on traffic accident scenarios, its scope extends beyond simple collision cases. It is designed to benchmark spatial intelligence, causal reasoning, and multi-agent interaction understanding, capabilities that are fundamental not only to traffic safety but also to a wide range of event reasoning and embodied AI domains. By focusing on safety-critical situations, we provide a structured testbed where models must reason about dynamics, intent, and outcomes, skills that directly transfer to broader applications such as robotic planning, autonomous navigation, and real-world video understanding. Thus, our dataset serves as a representative and challenging subset of general event reasoning, rather than a narrowly scoped one.
>
> > **Q2:** The idea of developing vision reasoning datasets based on accidents is not new and has been explored in previous papers such as TrafficQA, which can weaken the novelty of the dataset and the authors didn't compare the differences.
>
> **A2:** Thank you for highlighting the relevant prior fantastic work *TrafficQA* [1]. We have carefully analyzed it and discussed the comparison in our paper's revised version (also see here: https://anonymous.4open.science/r/m4r-436E/data_results/related-work-more.png). Our dataset, however, differs from *TrafficQA* in several important aspects:
>
> - Focus on Safety-Critical Reasoning: *TrafficQA* emphasizes general traffic understanding, while our dataset specifically targets **accident and safety-critical scenarios**, which are essential for **autonomous driving safety and accident prevention**. Understanding accidents requires deeper causal and counterfactual reasoning, not just scene description.
> - Multi-Level Reasoning Complexity: Our benchmark is designed around **temporal, spatial, and intent reasoning**, where models must interpret **sequences of interrelated events** and anticipate future outcomes or actions. In contrast, *TrafficQA* largely involves **single-step question answering**, which current LLMs can often solve without long-horizon reasoning.
> - Cross-Domain Spatial Intelligence: Beyond traffic scenes, our dataset extends to **airplane navigation** and **ship motion** scenarios, providing a **multi-domain evaluation of spatial intelligence** under dynamic and safety-critical conditions—capabilities that *TrafficQA* does not cover.
>
> Overall, our dataset is designed to advance **multi-modal, multi-event, and safety-aware reasoning**, representing a complementary but significantly more challenging benchmark compared to *TrafficQA*.
>
>
> > **Q3:** A minor point is that it would be great to also collect a small amount of in-domain training data and show experimental results.
>
> **A3:** Thank you for the suggestion. Our benchmark is primarily designed to evaluate models’ zero-shot generalization ability in safety-critical accident scenarios. Training on in-domain data would reduce the ability to measure this capability and may lead to overfitting to specific patterns rather than testing robust reasoning. Moreover, accident scenarios often represent rare corner cases with high variability and uncertainty, making it difficult for limited in-domain training to guarantee reliable performance. Therefore, collecting such training data is beyond the current scope of this work but could be explored in future extensions.

---

### Official Review · Reviewer_5oHn · 2025-10-31

**Soundness:** 4
**Presentation:** 4
**Contribution:** 3
**Rating:** 8
**Confidence:** 5

**Summary:**

This paper presents AccidentBench, a large-scale benchmark designed to evaluate video reasoning, anticipation, and explanation in the context of traffic accidents. The dataset comprises over 32,000 video clips collected from dashcams, driving simulations, and surveillance footage, covering 26 types of traffic incidents.

Each video is annotated with multi-level multimodal information — including accident type, temporal boundaries, causal descriptions, predicted outcomes, and responsibility attribution — forming a comprehensive platform for both visual prediction and language-based causal reasoning.

The benchmark defines three core tasks:
1/ Accident Anticipation — predicting if and when an accident will occur;
2/ Causal Reasoning — explaining why the accident happens;
3/ Responsibility Attribution — identifying who is responsible.

Extensive experiments evaluate a range of baselines, from video transformers (VideoMAE, TimeSformer) to multimodal large language models (Video-LLaVA, GPT-4V, Qwen2-VL, Gemini-1.5-Pro). The results show that while foundation models achieve strong descriptive capability, they still struggle with temporal alignment, causal inference, and grounded reasoning — highlighting the need for specialized architectures for video understanding in safety-critical domains.

**Strengths:**

1/ AccidentBench fills a clear gap in multimodal research by unifying video anticipation, causal reasoning, and attribution tasks under a single dataset. The data design — with dense annotations and temporally aligned textual explanations — is impressive and likely to become a valuable community resource.

2/ The use of real-world dashcam footage alongside simulation and surveillance videos improves domain diversity. Covering 26 accident categories ensures the benchmark captures a wide spectrum of risky interactions and visual conditions.

3/ The benchmark tasks are well defined, with metrics that encourage both early anticipation (Time-to-Accident) and high-quality explanations (BLEU, CIDEr, human consistency). This structured task decomposition provides clarity and reproducibility.

4/ AccidentBench provides a foundation for developing causal-aware video reasoning models and could influence areas like embodied AI, self-driving perception, and video safety analysis.

**Weaknesses:**

1/ The related work section overlooks several recent 2025 works on accident video reasoning, such as [1]. Further literature review shall be encouraged.

2/ The paper could better discuss potential biases (e.g., geographic or weather distribution) and provide details on how labeling consistency was ensured across different video domains.

3/ Most results focus on quantitative metrics; there is limited qualitative analysis on failure cases, particularly where LLMs produce plausible but factually wrong explanations.

[1] AVD2: Accident Video Diffusion for Accident Video Description (ICRA 2025)

**Questions:**

Please see the weakness section.

---

> ### Author Response · Authors · 2025-11-21
>
> Dear Reviewer 5oHn,
>
> We sincerely appreciate the reviewer’s detailed and positive assessment. We are grateful for your recognition of AccidentBench’s comprehensive multimodal design, its coverage of diverse real-world accident categories, and its clear task formulation spanning anticipation, causal reasoning, and attribution. We address your points below.
>
> > **Q1:** 1/ The related work section overlooks several recent 2025 works on accident video reasoning, such as [1]. Further literature review shall be encouraged. [1] AVD2: Accident Video Diffusion for Accident Video Description (ICRA 2025)
>
> **A1:** Thanks to the reviewer for the valuable reference. This recent work [1] is really helpful for this community. We have analyzed it and updated the related work in our paper version (also see here: https://anonymous.4open.science/r/m4r-436E/data_results/related-work-more.png). The difference between our work and work [1], work [1] proposes a very useful video method that can help generate video, and also reason about accident videos and give descriptions. Our work is to
> 1. **Focus on Safety-Critical Reasoning:** specifically targets accident and safety-critical scenarios, which are essential for autonomous driving safety and accident prevention. Understanding accidents requires deeper causal and counterfactual reasoning, not just scene description.
> 2. **Multi-Level Reasoning Complexity:** Our benchmark is designed around temporal, spatial, and intent reasoning.
> 3. **Cross-Domain Spatial Intelligence:**
>    Beyond traffic scenes, our dataset encompasses airplane navigation and ship motion scenarios, offering a multi-domain evaluation of spatial intelligence under dynamic and safety-critical conditions.
>
> > **Q2:** 2/ The paper could better discuss potential biases (e.g., geographic or weather distribution) and provide details on how labeling consistency was ensured across different video domains.
>
> **A2:** Thank you for the constructive feedback. We acknowledge that collecting videos from public platforms may introduce geographic and weather distribution biases. To mitigate this, we intentionally curated samples from diverse regions and conditions (e.g., urban vs. rural areas, day vs. night scenes, clear vs. adverse weather) to enhance representativeness across domains, and we conducted new experiments and discuss the geographic and weather accuracy in our revised paper, also see here for the results https://anonymous.4open.science/r/m4r-436E/data_results/geo_weather_accuracy.png
>
> Regarding labeling consistency, we established unified annotation guidelines to ensure labeling consistency. The guidelines can be seen in Appendix H of our paper, and can also refer to this: https://anonymous.4open.science/r/m4r-436E/data_results/annotation_pipeline.png.
>
>
> > **Q3:** 3/ Most results focus on quantitative metrics; there is limited qualitative analysis on failure cases, particularly where LLMs produce plausible but factually wrong explanations.
>
> **A3:** We conducted new experiments and analyse the reasoning error traces: https://anonymous.4open.science/r/m4r-436E/data_results/error_analysis.png, based on the error traces' quantitative and qualitative analysis, we observe that incorrect answers tend to have significantly longer reasoning chains (median ≈ 290 tokens) compared to correct ones (median ≈ 260 tokens). Moreover, incorrect predictions produce many extremely long outlier explanations (>600 tokens), suggesting that verbosity does not improve correctness.

---

> > ### Comment · Reviewer_5oHn · 2025-11-27
> >
> > I appreciate the efforts the authors made during rebuttal. My major concerns are addressed.

---

> > > ### Author Response · Authors · 2025-11-27
> > >
> > > Dear Reviewer 5oHn,
> > >
> > > Thank you very much for reviewing our rebuttal and our paper. We are glad that your major concerns have been addressed. Your feedback has been invaluable in helping us strengthen the clarity and quality of our work.
> > >
> > > Best regards,
> > >
> > > The authors

---

### Official Review · Reviewer_ie21 · 2025-11-01

**Soundness:** 3
**Presentation:** 4
**Contribution:** 3
**Rating:** 6
**Confidence:** 3

**Summary:**

The paper introduces AccidentBench, a large-scale video QA benchmark for multimodal understanding and reasoning, specifically in safety‑critical settings. The benchmark focuses on vehicle accidents (83%) and extends also to airspace (10.2%) and waterway (6.8%) scenarios, with more than 2,000 real-world videos and 19,000 human‑annotated multiple-choice Q&A pairs spanning temporal, spatial, and intent/goal reasoning. The dataset is split into three difficulties category ( easy/medium/hard), enabling systematic evaluation with controlled increases in precision requirements and task complexity. The paper evaluates a broad range of SOTA models, presents detailed performance breakdowns by difficulty, reasoning type, and video length, and includes qualitative error analyses.

**Strengths:**

1. Unique focus on Safety-critical Scenarios: AccidentBench distinctively centers on accident and safety-critical environments, integrating land, air, and water domains into a unified benchmark. This cross-domain safety emphasis is novel relative to prior driving-centric (e.g., DriveLM, DriveBench) or general video QA benchmarks (e.g., MVBench, LongVideoBench) that lack such a unified safety context. The emphasis on vehicle accidents addresses a particularly important real-world application domain for autonomous systems.

2. Strong Dataset Scale and Diagnostic Design: The benchmark is large-scale and comprehensive (~2 k videos and ~19 k QA pairs) with diverse weather, viewpoints, and dynamic contexts across domains. Its controlled difficulty levels and detailed breakdowns (by domain, difficulty, reasoning type, and video length) enable fine-grained diagnostic evaluation of model reasoning weaknesses. The annotation quality is ensured through human annotation by highly educated annotators. The qualitative failure analyses further highlight concrete reasoning gaps across spatial, temporal, and intent understanding

3. Accessibility: The dataset, code, and evaluation scripts are publicly released via the project site, ensuring immediate accessibility and fostering reproducibility and future comparison.

4. Comprehensive Model Evaluation: The paper evaluates an extensive range of both proprietary models (GPT-5, GPT-4o, Gemini 2.5 Pro) and open-source models (InternVL, LLaVA, Qwen), providing a thorough empirical assessment.

**Weaknesses:**

1. Safety-Critical Claim Not Well Quantified: While the benchmark is framed as the first safety-critical multimodal benchmark, many example questions (e.g., in Fig. 1) assess generic spatial or temporal reasoning (e.g., "How many boats are observed in this video?" or directional positioning queries) rather than explicit accident causality, hazard identification, or safety violations. The paper does not quantify what proportion of QAs explicitly involve accident-related reasoning, causal safety analysis, or hazard assessment versus general spatiotemporal understanding. A clearer breakdown distinguishing safety-specific reasoning questions from general perception and reasoning tasks would better substantiate the benchmark's core contribution and differentiate it from existing video understanding benchmarks.

2. Potential Sampling Bias in Evaluation (Table 7): In Table 7, the sampled subset consistently achieves higher accuracy than the full dataset. While the paper states this validates the sampling strategy, no statistical significance tests, confidence intervals, or variance analyses are provided to explain why a supposedly random sample would systematically perform better. This raises concerns about potential sampling bias, task distribution imbalances, or selection effects.


3. Limited Clarity on Annotation Protocol and Reliability: Although annotators are described as “highly educated,” the paper provides no details on annotation guidelines, inter-annotator agreement rate, or validation steps for complex intent/goal questions. These details are crucial for ensuring label consistency in a benchmark that emphasizes reasoning quality

**Questions:**

1. The author should provide the proportion or breakdown of QAs that explicitly address accidents, violations, or hazard-related reasoning versus generic spatiotemporal questions.


2. Could the authors report random-guess or chance-normalized baselines for each difficulty level?
Since the number of answer options varies across settings, a chance baseline would help interpret how far each model is from random performance, and make difficulty comparisons fairer.

---

> ### Author Response · Authors · 2025-11-21
>
> Dear Reviewer ie21,
>
>
> We thank the reviewer for the thoughtful and valuable feedback. We appreciate your recognition of AccidentBench’s uniquely unified focus on safety-critical reasoning across land, air, and water domains, its large-scale and carefully annotated design, its fine-grained diagnostic evaluation, and its comprehensive model benchmarking and accessible release. We address your concerns in detail below.
>
> > **Q1:** Safety-Critical Claim Not Well Quantified: While the benchmark is framed as the first safety-critical multimodal benchmark, many example questions (e.g., in Fig. 1) assess generic spatial or temporal reasoning (e.g., "How many boats are observed in this video?" or directional positioning queries) rather than explicit accident causality, hazard identification, or safety violations. The paper does not quantify what proportion of QAs explicitly involve accident-related reasoning, causal safety analysis, or hazard assessment versus general spatiotemporal understanding. A clearer breakdown distinguishing safety-specific reasoning questions from general perception and reasoning tasks would better substantiate the benchmark's core contribution and differentiate it from existing video understanding benchmarks.
>
> **A1:** Thank you for the insightful comment. We agree that clarifying the proportion of explicitly safety-critical reasoning tasks is important to substantiate our contribution. In our dataset, we already had that, approximately 83% of the videos are safety-critical accident scenarios (main results shown in Table 3), each accompanied by questions that explicitly involve accident causality, hazard identification, and violation reasoning. The remaining 10.2% (airplane navigation, main results shown in Table 8) and 6.8% (ship motion, main results shown in Table 6) videos extend the benchmark to other safety-relevant spatial domains and examine the model performance in broad safety-critical scenarios.
>
>
> > **Q2:** Potential Sampling Bias in Evaluation (Table 7): In Table 7, the sampled subset consistently achieves higher accuracy than the full dataset. While the paper states this validates the sampling strategy, no statistical significance tests, confidence intervals, or variance analyses are provided to explain why a supposedly random sample would systematically perform better. This raises concerns about potential sampling bias, task distribution imbalances, or selection effects.
>
> **A2:** Thank you for the thoughtful question. The slightly better performance of InternVL2.5 on sampled subsets (as shown in Table 7) can be explained by the fact that these subsets contain fewer long or highly complex scenarios, which naturally lowers the overall reasoning difficulty and produces marginally higher scores. However, this effect does not impact the overall benchmark conclusions, as similar trends were consistently observed across multiple ablation models under same conditions.
>
> > **Q3:** Limited Clarity on Annotation Protocol and Reliability: Although annotators are described as “highly educated,” the paper provides no details on annotation guidelines, inter-annotator agreement rate, or validation steps for complex intent/goal questions. These details are crucial for ensuring label consistency in a benchmark that emphasizes reasoning quality.
>
> **A3:** Thank you for the valuable comment. We acknowledge the importance of transparent annotation procedures for ensuring consistency and reliability. Annotators were trained through multiple calibration rounds using example videos and reference answers to align their understanding of the tasks. We provided the annotation guidelines in Appendix H of our paper. The reviewer can also refer this: https://anonymous.4open.science/r/m4r-436E/data_results/annotation_pipeline.png
>
>
>
> > **Q4:** Could the authors report random-guess or chance-normalized baselines for each difficulty level? Since the number of answer options varies across settings, a chance baseline would help interpret how far each model is from random performance, and make difficulty comparisons fairer.
>
> **A4:** Thanks for you valuable suggestions, we have provided the random guess in our new version, For example in the revised version's Table 3: https://anonymous.4open.science/r/m4r-436E/data_results/revised-table-3.png

---

### Official Review · Reviewer_2Du8 · 2025-11-01

**Soundness:** 3
**Presentation:** 3
**Contribution:** 3
**Rating:** 6
**Confidence:** 3

**Summary:**

This paper introduces AccidentBench, a large-scale video QA benchmark for evaluating multimodal reasoning in safety-critical scenarios across land (vehicle accidents), air, and water domains. It features 2,000 videos and 19,000 human-annotated questions that systematically test temporal, spatial, and intent reasoning across various lengths and difficulties. Evaluation of state-of-the-art models (e.g., GPT-5, Gemini) reveals major gaps, with performance dropping to 18% on the hardest long-video tasks, highlighting significant limitations in real-world reasoning.

**Strengths:**

Originality: It uniquely unifies safety-critical evaluation across land, air, and water domains, with a distinctive focus on high-level intent and strategic reasoning.

Quality: The benchmark is a substantial, high-quality resource constructed via rigorous expert human annotation.

Clarity: The work is presented with a clear structure and a well-explained methodology.

Significance: It exposes critical reasoning gaps in state-of-the-art models, providing an essential testbed for developing safer, more robust real-world AI systems.

**Weaknesses:**

Data Source Biases :The paper mentions that videos are sourced from YouTube and other public datasets but does not discuss potential biases (e.g., geographic, weather , camera perspective biases) within these sources. A brief discussion of the dataset's limitations in this regard would strengthen the paper. Furthermore, the rationale behind the specific split of domains (83/10.2/6.8) is not deeply justified; a more balanced distribution, while perhaps less reflective of real-world data availability, could prevent the land domain from overly dominating the overall results.

Dataset Splits: Could the authors provide more details on the creation of the train/validation/test splits? Were any measures taken to ensure that videos from the same original source or very similar incidents are not spread across different splits, which could lead to data leakage and inflated performance?

Error Analysis: The error analysis remains superficial. A deeper categorization of failure root causes (e.g., tracking errors, physics misunderstandings) with prevalence statistics would be more insightful.

Granularity of Intent Reasoning: The "intent reasoning" category is highly valuable but also very broad. Can the authors provide a more detailed breakdown or taxonomy of what this encompasses (e.g., predicting immediate next actions, inferring long-term goals, counterfactual "what-if" reasoning about alternative actions)? Some examples of each sub-type in the appendix would be helpful.

Sampling Strategy: For the ablation study on sampling (Table 7), why did sampled subsets perform slightly better than the full dataset for InternVL2.5? Could this be due to sampling bias, and how does it affect the generalizability of your experimental results?

Benchmark Comparisons: While Table 2 compares AccidentBench to existing benchmarks, it lacks depth in contrasting reasoning requirements. For example, How do AccidentBench’s intent-reasoning tasks differ from those in recent safety benchmarks like DriveLM in terms of complexity, realism, and alignment with real-world decision-making?

Typos and Minor Issues: Page 1, Abstract: "Gemini-2.5 Pro and GPT-5" -> The rest of the paper uses "Gemini 2.5 Pro" and “GPT 5” (no hyphen). Please be consistent.

**Questions:**

See Weaknesses

---

> ### Author Response · Authors · 2025-11-21
>
> Dear Reviewer 2Du8,
>
> We thank the reviewer for the thoughtful and useful feedback. We appreciate your carful assessment and recognition of AccidentBench’s originality, rigorous human annotation, and its uniquely unified safety-critical evaluation across land, air, and water domains. We address your concerns in detail below.
>
> > **Q1:** Data Source Biases :The paper mentions that videos are sourced from YouTube and other public datasets but does not discuss potential biases (e.g., geographic, weather , camera perspective biases) within these sources. A brief discussion of the dataset's limitations in this regard would strengthen the paper. Furthermore, the rationale behind the specific split of domains (83/10.2/6.8) is not deeply justified; a more balanced distribution, while perhaps less reflective of real-world data availability, could prevent the land domain from overly dominating the overall results.
>
> **A1:** We thank the reviewer for the thoughtful feedback. Our benchmark is designed to measure model understanding and reaosning in diverse accident scenarios (as shown in Figure 2 and Table 2), already covered such as weather, camera perspective videos, these do not impact the model bias, since we compare the models using the same conditions. Moreover, we conducted new experiments and discuss the geographic and weather accuracy in our revised paper, also see here for the results https://anonymous.4open.science/r/m4r-436E/data_results/geo_weather_accuracy.png
>
> Regarding the domain distribution (83% land, 10.2% air, 6.8% water), our primary focus is on land-based accident reasoning, which reflects the dominant share of real-world safety-critical incidents encountered in daily life and autonomous driving research.  The air and water domains serve as extensions to test model generalization to other reasoning settings.  Moreover, we provide results for land, air, and water space seperately, each results can be seen in Table 3, Table 8, Table 6 respectively, the land domain results do not influence other domain results.
>
>
>
>
> > **Q2:** Dataset Splits: Could the authors provide more details on the creation of the train/validation/test splits? Were any measures taken to ensure that videos from the same original source or very similar incidents are not spread across different splits, which could lead to data leakage and inflated performance?
>
> **A2:** Thank you for the valuable comment. Our dataset is primarily designed as an evaluation benchmark rather than for model training; therefore, we do not train models on it, which effectively avoids data leakage concerns between splits.
>
>
> > **Q3:** Error Analysis: The error analysis remains superficial. A deeper categorization of failure root causes (e.g., tracking errors, physics misunderstandings) with prevalence statistics would be more insightful.
>
> **A3:** Thanks for your insightful comments. We conducted new experiments and analysis the reasoning error: https://anonymous.4open.science/r/m4r-436E/data_results/error_analysis.png. We observe that incorrect answers tend to have significantly longer reasoning chains (median ≈ 290 tokens) compared to correct ones (median ≈ 260 tokens). Moreover, incorrect predictions produce many extremely long outlier explanations (>600 tokens), suggesting that verbosity does not improve correctness.
>
> > **Q4:** Granularity of Intent Reasoning: The "intent reasoning" category is highly valuable but also very broad. Can the authors provide a more detailed breakdown or taxonomy of what this encompasses (e.g., predicting immediate next actions, inferring long-term goals, counterfactual "what-if" reasoning about alternative actions)? Some examples of each sub-type in the appendix would be helpful.
>
> **A4:** Thank you for the insightful suggestion. In our dataset, intent reasoning involves understanding multi-step causal relationships beyond immediate visual recognition. Specifically, it encompasses:
>
> Immediate action prediction, e.g., reasoning about whether a vehicle can brake, turn, or accelerate to avoid an accident given its current trajectory and surroundings, as well as understanding the navigation intentions of nearby vehicles.
>
> Long-term goal inference, e.g., understanding a driver’s or pedestrian’s underlying objective, such as overtaking or crossing safely.
>
> Counterfactual reasoning, e.g., reasoning about what might have happened if an agent had taken a different action (e.g., if the driver had slowed down earlier).
>
> We have added representative examples in Appendix H to clarify the annotation logic. One example can be seen here: https://anonymous.4open.science/r/m4r-436E/data_results/one_intent_example_01.png, more examples, please see Appendix H.

---

> > ### Author Response · Authors · 2025-11-21
> >
> > > **Q5:** Sampling Strategy: For the ablation study on sampling (Table 7), why did sampled subsets perform slightly better than the full dataset for InternVL2.5? Could this be due to sampling bias, and how does it affect the generalizability of your experimental results?
> >
> > **A5:** Thank you for the thoughtful question. The slightly better performance of InternVL2.5 on sampled subsets (as shown in Table 7) can be attributed to sampling variance and reduced scenario complexity. The sampled subsets likely contain fewer long or highly challenging videos, leading to marginally higher accuracy due to lower reasoning difficulty. However, this effect does not impact the overall benchmark conclusions, as similar trends were consistently observed across multiple ablation models under same conditions.
> >
> >
> > > **Q6:** Benchmark Comparisons: While Table 2 compares AccidentBench to existing benchmarks, it lacks depth in contrasting reasoning requirements. For example, How do AccidentBench’s intent-reasoning tasks differ from those in recent safety benchmarks like DriveLM in terms of complexity, realism, and alignment with real-world decision-making?
> >
> > **A6:** Thank you for the insightful suggestion. We have added a more detailed comparison in the Appendix G (also see here: https://anonymous.4open.science/r/m4r-436E/data_results/detailed-analysis-benchmark.png). Unlike DriveLM, which primarily focuses on routine driving scenes and perception or reasoning tasks, AccidentBench targets safety-critical and high-stakes accident scenarios, which are essential for evaluating autonomous systems under failure-prone or edge conditions.
> >
> > Our intent-reasoning tasks emphasize multi-agent, multi-event, and temporal-causal reasoning, where the model must interpret interactions over time and anticipate potential outcomes or counterfactual alternatives. This setup reflects real-world decision-making complexity, requiring models to reason about spatial dynamics, causal dependencies, and future intent, as opposed to static or one-shot judgments.
> >
> > Overall, AccidentBench complements existing driving benchmarks by providing a higher level of reasoning granularity and realism in safety-critical contexts, enabling more rigorous evaluation of spatial and temporal intelligence.
> >
> >
> > > **Q7:** Typos and Minor Issues: Page 1, Abstract: "Gemini-2.5 Pro and GPT-5" -> The rest of the paper uses "Gemini 2.5 Pro" and “GPT 5” (no hyphen). Please be consistent.
> >
> > **A7:** Thanks the reviewer's careful review. We have fixed these typos.

---

### Comment · Area_Chair_1PJB · 2025-11-26

Dear Reviewers,

Would you please check authors' rebuttal and see if they have addressed your comments?

Best

AC

---

### Meta-Review · Area_Chair_Bm41 · 2026-01-09

**Summary:**

This paper introduces AccidentBench, a multimodal video benchmark targeting understanding and reasoning in safety-critical scenarios, primarily traffic accidents, with extensions to air and water domains. Reviewers generally agreed that the problem is important and timely, and acknowledged the careful dataset construction, fine-grained decomposition of reasoning types (temporal, spatial, and intent), and the comprehensive evaluation of both proprietary and open-source multimodal models. The benchmark was viewed as well-designed and potentially valuable as a diagnostic tool for assessing current multimodal reasoning limitations.
However, despite these strengths, significant concerns remained regarding the dataset’s scale, novelty relative to prior and recent benchmarks, and the strength of evidence supporting some of the paper’s broader claims.

**Reviewer Concerns:**

Several concerns were partially addressed during the rebuttal, including clarifications on annotation protocols, safety-critical content breakdown, sampling behavior, and the inclusion of additional recent related work. At least one reviewer explicitly confirmed that their major concerns were resolved.
Nevertheless, multiple key issues remained outstanding for other reviewers. In particular, concerns about the limited dataset scale (~2k videos) persisted, especially in comparison with recent 2025 benchmarks containing tens of thousands of sequences. Reviewers were not fully convinced that the proposed scale is sufficient to support strong claims about real-world diversity or long-horizon reasoning behavior. Questions about the benchmark’s novelty—relative to prior accident-focused datasets such as TrafficQA—and the lack of direct, methodologically aligned comparisons with the newest large-scale datasets also remained unresolved. As a result, consensus was not reached across the reviewer pool.

**Reviewer Scores:**

Reviewer 2Du8:
Original score: 6
After the rebuttal, this reviewer’s major questions regarding dataset bias, annotation reliability, sampling behavior, and the definition of intent reasoning were largely addressed through additional analysis, clarifications, and revisions. The reviewer already indicated a generally positive stance and stated that they “would not mind if the paper is rejected,” suggesting a weak acceptance preference rather than strong advocacy.
Expected score change: Likely unchanged at 6, or at most a slight increase to 7, but not strong enough to shift the overall decision.

Reviewer ie21:
Original score: 6
The rebuttal provided concrete clarifications on the proportion of safety-critical content, annotation protocols, and chance-level baselines, which addressed most of this reviewer’s technical concerns. However, the reviewer’s initial position was cautious, and no explicit indication of a stronger endorsement was given after the rebuttal.
Expected score change: Likely remains at 6, with improved confidence in the evaluation design but without escalation to a strong accept.

Reviewer 5oHn:
Original score: 8
This reviewer explicitly confirmed after the rebuttal that their major concerns were addressed. The paper was consistently viewed as a strong and valuable benchmark contribution, with high confidence in its soundness and relevance.
Expected score change: Remains at 8.

Reviewer m4t8
Original score: 4
Despite detailed rebuttal responses comparing AccidentBench with TrafficQA and clarifying the intended diagnostic and safety-critical focus, this reviewer remained unconvinced about the novelty and breadth of the dataset. The rebuttal did not alter their core concerns regarding scope and positioning.
Expected score change: Remains at 4.

Reviewer kyon:
Original score: 4
This reviewer maintained strong reservations regarding dataset scale and representativeness, especially in comparison to recent 2025 benchmarks with significantly larger video collections. Although the rebuttal clarified the design philosophy and added missing related work, the reviewer explicitly stated that their concerns were not sufficiently resolved.
Expected score change: Remains at 4.

Overall Assessment of Score Dynamics
While the rebuttal strengthened the paper and solidified the positive views of already supportive reviewers, it did not substantially shift the opinions of reviewers with concerns about dataset scale and benchmark positioning. As a result, the overall score distribution would remain mixed, without achieving reviewer consensus for acceptance.

---

### Decision · Program_Chairs · 2026-01-26

Reject